# GANs Trained by a Two Time-Scale Update Rule Converge to a Local Nash Equilibrium

**Martin Heusel**　　**Hubert Ramsauer**　　**Thomas Unterthiner**　　**Bernhard Nessler**

**Sepp Hochreiter**

LIT AI Lab & Institute of Bioinformatics,
Johannes Kepler University Linz
A-4040 Linz, Austria
{mhe,ramsauer,unterthiner,nessler,hochreit}@bioinf.jku.at

## Abstract

Generative Adversarial Networks (GANs) excel at creating realistic images with complex models for which maximum likelihood is infeasible. However, the convergence of GAN training has still not been proved. We propose a two time-scale update rule (TTUR) for training GANs with stochastic gradient descent on arbitrary GAN loss functions. TTUR has an individual learning rate for both the discriminator and the generator. Using the theory of stochastic approximation, we prove that the TTUR converges under mild assumptions to a stationary local Nash equilibrium. The convergence carries over to the popular Adam optimization, for which we prove that it follows the dynamics of a heavy ball with friction and thus prefers flat minima in the objective landscape. For the evaluation of the performance of GANs at image generation, we introduce the 'Fréchet Inception Distance" (FID) which captures the similarity of generated images to real ones better than the Inception Score. In experiments, TTUR improves learning for DCGANs and Improved Wasserstein GANs (WGAN-GP) outperforming conventional GAN training on CelebA, CIFAR-10, SVHN, LSUN Bedrooms, and the One Billion Word Benchmark.

## 1 Introduction

Generative adversarial networks (GANs) [16] have achieved outstanding results in generating realistic images [42, 31, 25, 1, 4] and producing text [21]. GANs can learn complex generative models for which maximum likelihood or a variational approximations are infeasible. Instead of the likelihood, a discriminator network serves as objective for the generative model, that is, the generator. GAN learning is a game between the generator, which constructs synthetic data from random variables, and the discriminator, which separates synthetic data from real world data. The generator's goal is to construct data in such a way that the discriminator cannot tell them apart from real world data. Thus, the discriminator tries to minimize the synthetic-real discrimination error while the generator tries to maximize this error. Since training GANs is a game and its solution is a Nash equilibrium, gradient descent may fail to converge [44, 16, 18]. Only *local* Nash equilibria are found, because gradient descent is a local optimization method. If there exists a local neighborhood around a point in parameter space where neither the generator nor the discriminator can unilaterally decrease their respective losses, then we call this point a local Nash equilibrium.

To characterize the convergence properties of training general GANs is still an open challenge [17, 18]. For special GAN variants, convergence can be proved under certain assumptions [34, 20, 46]. A

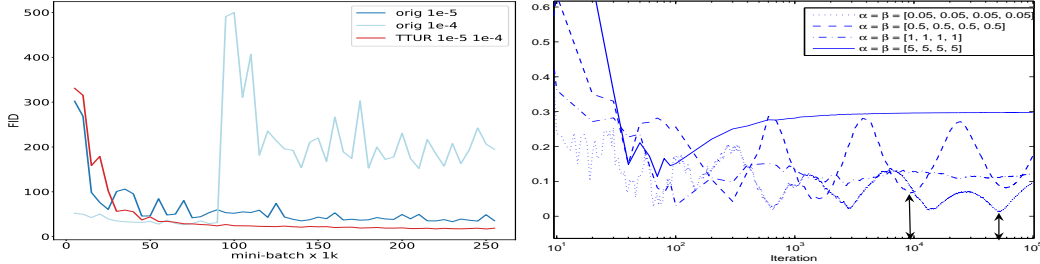

Figure 1: Left: Original vs. TTUR GAN training on CelebA. Right: Figure from Zhang 2007 [50] which shows the distance of the parameter from the optimum for a one time-scale update of a 4 node network flow problem. When the upper bounds on the errors $(\alpha, \beta)$ are small, the iterates oscillate and repeatedly return to a neighborhood of the optimal solution (cf. Supplement Section 2.3). However, when the upper bounds on the errors are large, the iterates typically diverge.

prerequisit for many convergence proofs is local stability [30] which was shown for GANs by Nagarajan and Kolter [39] for a min-max GAN setting. However, Nagarajan and Kolter require for their proof either rather strong and unrealistic assumptions or a restriction to a linear discriminator. Recent convergence proofs for GANs hold for expectations over training samples or for the number of examples going to infinity [32, 38, 35, 2], thus do not consider mini-batch learning which leads to a stochastic gradient [47, 23, 36, 33].

Recently actor-critic learning has been analyzed using stochastic approximation. Prasad et al. [41] showed that a two time-scale update rule ensures that training reaches a stationary local Nash equilibrium if the critic learns faster than the actor. Convergence was proved via an ordinary differential equation (ODE), whose stable limit points coincide with stationary local Nash equilibria. We follow the same approach. We adopt this approach for GANs and prove that also GANs converge to a local Nash equilibrium when trained by a two time-scale update rule (TTUR), i.e., when discriminator and generator have separate learning rates. This also leads to better results in experiments. The main premise is that the discriminator converges to a local minimum when the generator is fixed. If the generator changes slowly enough, then the discriminator still converges, since the generator perturbations are small. Besides ensuring convergence, the performance may also improve since the discriminator must first learn new patterns before they are transferred to the generator. In contrast, a generator which is overly fast, drives the discriminator steadily into new regions without capturing its gathered information. In recent GAN implementations, the discriminator often learned faster than the generator. A new objective slowed down the generator to prevent it from overtraining on the current discriminator [44]. The Wasserstein GAN algorithm uses more update steps for the discriminator than for the generator [1]. We compare TTUR and standard GAN training. Fig. 1 shows at the left panel a stochastic gradient example on CelebA for original GAN training (orig), which often leads to oscillations, and the TTUR. On the right panel an example of a 4 node network flow problem of Zhang et al. [50] is shown. The distance between the actual parameter and its optimum for an one time-scale update rule is shown across iterates. When the upper bounds on the errors are small, the iterates return to a neighborhood of the optimal solution, while for large errors the iterates may diverge (see also Supplement Section 2.3). Our novel contributions in this paper are: (i) the two time-scale update rule for GANs, (ii) the proof that GANs trained with TTUR converge to a stationary local Nash equilibrium, (iii) the description of Adam as heavy ball with friction and the resulting second order differential equation, (iv) the convergence of GANs trained with TTUR and Adam to a stationary local Nash equilibrium, (v) the "Fréchet Inception Distance" (FID) to evaluate GANs, which is more consistent than the Inception Score.

## Two Time-Scale Update Rule for GANs

We consider a discriminator $D(.; \boldsymbol{w})$ with parameter vector $\boldsymbol{w}$ and a generator $G(.; \boldsymbol{\theta})$ with parameter vector $\boldsymbol{\theta}$. Learning is based on a stochastic gradient $\tilde{\boldsymbol{g}}(\boldsymbol{\theta}, \boldsymbol{w})$ of the discriminator's loss function $\mathcal{L}_D$ and a stochastic gradient $\tilde{\boldsymbol{h}}(\boldsymbol{\theta}, \boldsymbol{w})$ of the generator's loss function $\mathcal{L}_G$. The loss functions $\mathcal{L}_D$ and $\mathcal{L}_G$ can be the original as introduced in Goodfellow et al. [16], its improved versions [18], or recently proposed losses for GANs like the Wasserstein GAN [1]. Our setting is not restricted to min-max

GANs, but is also valid for all other, more general GANs for which the discriminator's loss function $\mathcal{L}_D$ is not necessarily related to the generator's loss function $\mathcal{L}_G$. The gradients $\tilde{g}(\boldsymbol{\theta}, \boldsymbol{w})$ and $\tilde{h}(\boldsymbol{\theta}, \boldsymbol{w})$ are stochastic, since they use mini-batches of $m$ real world samples $\boldsymbol{x}^{(i)}, 1 \leqslant i \leqslant m$ and $m$ synthetic samples $\boldsymbol{z}^{(i)}, 1 \leqslant i \leqslant m$ which are randomly chosen. If the true gradients are $\boldsymbol{g}(\boldsymbol{\theta}, \boldsymbol{w}) = \nabla_w \mathcal{L}_D$ and $\boldsymbol{h}(\boldsymbol{\theta}, \boldsymbol{w}) = \nabla_\theta \mathcal{L}_G$, then we can define $\tilde{g}(\boldsymbol{\theta}, \boldsymbol{w}) = \boldsymbol{g}(\boldsymbol{\theta}, \boldsymbol{w}) + \boldsymbol{M}^{(w)}$ and $\tilde{h}(\boldsymbol{\theta}, \boldsymbol{w}) = \boldsymbol{h}(\boldsymbol{\theta}, \boldsymbol{w}) + \boldsymbol{M}^{(\theta)}$ with random variables $\boldsymbol{M}^{(w)}$ and $\boldsymbol{M}^{(\theta)}$. Thus, the gradients $\tilde{g}(\boldsymbol{\theta}, \boldsymbol{w})$ and $\tilde{h}(\boldsymbol{\theta}, \boldsymbol{w})$ are stochastic approximations to the true gradients. Consequently, we analyze convergence of GANs by two time-scale stochastic approximations algorithms. For a two time-scale update rule (TTUR), we use the learning rates $b(n)$ and $a(n)$ for the discriminator and the generator update, respectively:

$$\boldsymbol{w}_{n+1} = \boldsymbol{w}_n + b(n) \left( \boldsymbol{g}(\boldsymbol{\theta}_n, \boldsymbol{w}_n) + \boldsymbol{M}_n^{(w)} \right), \ \boldsymbol{\theta}_{n+1} = \boldsymbol{\theta}_n + a(n) \left( \boldsymbol{h}(\boldsymbol{\theta}_n, \boldsymbol{w}_n) + \boldsymbol{M}_n^{(\theta)} \right). \quad (1)$$

For more details on the following convergence proof and its assumptions see Supplement Section 2.1. To prove convergence of GANs learned by TTUR, we make the following assumptions (The actual assumption is ended by ◄, the following text are just comments and explanations):

(A1) The gradients $\boldsymbol{h}$ and $\boldsymbol{g}$ are Lipschitz. ◄ Consequently, networks with Lipschitz smooth activation functions like ELUs ($\alpha = 1$) [11] fulfill the assumption but not ReLU networks.

(A2) $\sum_n a(n) = \infty, \sum_n a^2(n) < \infty, \sum_n b(n) = \infty, \sum_n b^2(n) < \infty, a(n) = \mathrm{o}(b(n))$◄

(A3) The stochastic gradient errors $\{\boldsymbol{M}_n^{(\theta)}\}$ and $\{\boldsymbol{M}_n^{(w)}\}$ are martingale difference sequences w.r.t. the increasing $\sigma$-field $\mathcal{F}_n = \sigma(\boldsymbol{\theta}_l, \boldsymbol{w}_l, \boldsymbol{M}_l^{(\theta)}, \boldsymbol{M}_l^{(w)}, l \leqslant n), n \geqslant 0$ with $\mathrm{E}\left[ \|\boldsymbol{M}_n^{(\theta)}\|^2 \mid \mathcal{F}_n^{(\theta)} \right] \leqslant B_1$ and $\mathrm{E}\left[ \|\boldsymbol{M}_n^{(w)}\|^2 \mid \mathcal{F}_n^{(w)} \right] \leqslant B_2$, where $B_1$ and $B_2$ are positive deterministic constants.◄ The original Assumption (A3) from Borkar 1997 follows from Lemma 2 in [5] (see also [43]). The assumption is fulfilled in the Robbins-Monro setting, where mini-batches are randomly sampled and the gradients are bounded.

(A4) For each $\boldsymbol{\theta}$, the ODE $\dot{\boldsymbol{w}}(t) = \boldsymbol{g}(\boldsymbol{\theta}, \boldsymbol{w}(t))$ has a local asymptotically stable attractor $\boldsymbol{\lambda}(\boldsymbol{\theta})$ within a domain of attraction $G_\theta$ such that $\boldsymbol{\lambda}$ is Lipschitz. The ODE $\dot{\boldsymbol{\theta}}(t) = \boldsymbol{h}(\boldsymbol{\theta}(t), \boldsymbol{\lambda}(\boldsymbol{\theta}(t)))$ has a local asymptotically stable attractor $\boldsymbol{\theta}^*$ within a domain of attraction.◄ The discriminator must converge to a minimum for fixed generator parameters and the generator, in turn, must converge to a minimum for this fixed discriminator minimum. Borkar 1997 required unique global asymptotically stable equilibria [7]. The assumption of global attractors was relaxed to local attractors via Assumption (A6) and Theorem 2.7 in Karmakar & Bhatnagar [26]. See for more details Assumption (A6) in Supplement Section 2.1.3. Here, the GAN objectives may serve as Lyapunov functions. These assumptions of locally stable ODEs can be ensured by an additional weight decay term in the loss function which increases the eigenvalues of the Hessian. Therefore, problems with a region-wise constant discriminator that has zero second order derivatives are avoided. For further discussion see Supplement Section 2.1.1 (C3).

(A5) $\sup_n \|\boldsymbol{\theta}_n\| < \infty$ and $\sup_n \|\boldsymbol{w}_n\| < \infty$.◄ Typically ensured by the objective or a weight decay term.

The next theorem has been proved in the seminal paper of Borkar 1997 [7].

**Theorem 1** (Borkar). *If the assumptions are satisfied, then the updates Eq.* (1) *converge to* $(\boldsymbol{\theta}^*, \boldsymbol{\lambda}(\boldsymbol{\theta}^*))$ *a.s.*

The solution $(\boldsymbol{\theta}^*, \boldsymbol{\lambda}(\boldsymbol{\theta}^*))$ is a stationary local Nash equilibrium [41], since $\boldsymbol{\theta}^*$ as well as $\boldsymbol{\lambda}(\boldsymbol{\theta}^*)$ are local asymptotically stable attractors with $\boldsymbol{g}(\boldsymbol{\theta}^*, \boldsymbol{\lambda}(\boldsymbol{\theta}^*)) = \boldsymbol{0}$ and $\boldsymbol{h}(\boldsymbol{\theta}^*, \boldsymbol{\lambda}(\boldsymbol{\theta}^*)) = \boldsymbol{0}$. An alternative approach to the proof of convergence using the Poisson equation for ensuring a solution to the fast update rule can be found in the Supplement Section 2.1.2. This approach assumes a linear update function in the fast update rule which, however, can be a linear approximation to a nonlinear gradient [28, 29]. For the rate of convergence see Supplement Section 2.2, where Section 2.2.1 focuses on linear and Section 2.2.2 on non-linear updates. For equal time-scales it can only be proven that the updates revisit an environment of the solution infinitely often, which, however, can be very large [50, 12]. For more details on the analysis of equal time-scales see Supplement Section 2.3. The main idea of the proof of Borkar [7] is to use $(T, \delta)$ perturbed ODEs according to Hirsch 1989 [22] (see also Appendix Section C of Bhatnagar, Prasad, & Prashanth 2013 [6]). The proof relies on the fact

that there eventually is a time point when the perturbation of the slow update rule is small enough (given by $\delta$) to allow the fast update rule to converge. For experiments with TTUR, we aim at finding learning rates such that the slow update is small enough to allow the fast to converge. Typically, the slow update is the generator and the fast update the discriminator. We have to adjust the two learning rates such that the generator does not affect discriminator learning in a undesired way and perturb it too much. However, even a larger learning rate for the generator than for the discriminator may ensure that the discriminator has low perturbations. Learning rates cannot be translated directly into perturbation since the perturbation of the discriminator by the generator is different from the perturbation of the generator by the discriminator.

## 2  Adam Follows an HBF ODE and Ensures TTUR Convergence

In our experiments, we aim at using Adam stochastic approximation to avoid mode collapsing. GANs suffer from "mode collapsing" where large masses of probability are mapped onto a few modes that cover only small regions. While these regions represent meaningful samples, the variety of the real world data is lost and only few prototype samples are generated. Different methods have been proposed to avoid mode collapsing [9, 37]. We obviate mode collapsing by using Adam stochastic approximation [27]. Adam can be described as Heavy Ball with Friction (HBF) (see below), since it averages over past gradients. This averaging corresponds to a velocity that makes the generator resistant to getting pushed into small regions. Adam as an HBF method typically overshoots small local minima that correspond to mode collapse and can find flat minima which generalize well [24]. Fig. 2 depicts the dynamics of HBF, where the ball settles at a flat minimum. Next, we analyze whether GANs trained with TTUR converge when using Adam. For more details see Supplement Section 3.

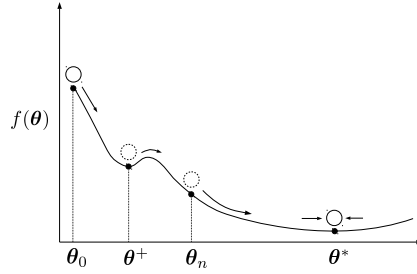

**Figure 2:** Heavy Ball with Friction, where the ball with mass overshoots the local minimum $\boldsymbol{\theta}^+$ and settles at the flat minimum $\boldsymbol{\theta}^*$.

We recapitulate the Adam update rule at step $n$, with learning rate $a$, exponential averaging factors $\beta_1$ for the first and $\beta_2$ for the second moment of the gradient $\nabla f(\boldsymbol{\theta}_{n-1})$:

$$
\begin{aligned}
\boldsymbol{g}_n &\longleftarrow \nabla f(\boldsymbol{\theta}_{n-1}) \\
\boldsymbol{m}_n &\longleftarrow (\beta_1/(1-\beta_1^n))\,\boldsymbol{m}_{n-1} + ((1-\beta_1)/(1-\beta_1^n))\,\boldsymbol{g}_n \\
\boldsymbol{v}_n &\longleftarrow (\beta_2/(1-\beta_2^n))\,\boldsymbol{v}_{n-1} + ((1-\beta_2)/(1-\beta_2^n))\,\boldsymbol{g}_n \odot \boldsymbol{g}_n \\
\boldsymbol{\theta}_n &\longleftarrow \boldsymbol{\theta}_{n-1} - a\,\boldsymbol{m}_n/(\sqrt{\boldsymbol{v}_n}+\epsilon)\,,
\end{aligned} \tag{2}
$$

where following operations are meant componentwise: the product $\odot$, the square root $\sqrt{\cdot}$, and the division / in the last line. Instead of learning rate $a$, we introduce the damping coefficient $a(n)$ with $a(n) = an^{-\tau}$ for $\tau \in (0, 1]$. Adam has parameters $\beta_1$ for averaging the gradient and $\beta_2$ parametrized by a positive $\alpha$ for averaging the squared gradient. These parameters can be considered as defining a memory for Adam. To characterize $\beta_1$ and $\beta_2$ in the following, we define the exponential memory $r(n) = r$ and the polynomial memory $r(n) = r/\sum_{l=1}^{n} a(l)$ for some positive constant $r$. The next theorem describes Adam by a differential equation, which in turn allows to apply the idea of $(T, \delta)$ perturbed ODEs to TTUR. Consequently, learning GANs with TTUR and Adam converges.

**Theorem 2.** *If Adam is used with $\beta_1 = 1 - a(n+1)r(n)$, $\beta_2 = 1 - \alpha a(n+1)r(n)$ and with $\nabla f$ as the full gradient of the lower bounded, continuously differentiable objective $f$, then for stationary second moments of the gradient, Adam follows the differential equation for Heavy Ball with Friction (HBF):*

$$
\ddot{\boldsymbol{\theta}}_t + a(t)\,\dot{\boldsymbol{\theta}}_t + \nabla f(\boldsymbol{\theta}_t) = \boldsymbol{0}\,. \tag{3}
$$

*Adam converges for gradients $\nabla f$ that are L-Lipschitz.*

*Proof.* Gadat et al. derived a discrete and stochastic version of Polyak's Heavy Ball method [40], the Heavy Ball with Friction (HBF) [15]:

$$
\begin{aligned}
\boldsymbol{\theta}_{n+1} &= \boldsymbol{\theta}_n - a(n+1)\,\boldsymbol{m}_n\,, \\
\boldsymbol{m}_{n+1} &= \big(1 - a(n+1)\,r(n)\big)\,\boldsymbol{m}_n + a(n+1)\,r(n)\,\big(\nabla f(\boldsymbol{\theta}_n) + \boldsymbol{M}_{n+1}\big)\,.
\end{aligned} \tag{4}
$$

These update rules are the first moment update rules of Adam [27]. The HBF can be formulated as the differential equation Eq. (3) [15]. Gadat et al. showed that the update rules Eq. (4) converge for loss functions $f$ with at most quadratic grow and stated that convergence can be proofed for $\nabla f$ that are $L$-Lipschitz [15]. Convergence has been proved for continuously differentiable $f$ that is quasiconvex (Theorem 3 in Goudou & Munier [19]). Convergence has been proved for $\nabla f$ that is $L$-Lipschitz and bounded from below (Theorem 3.1 in Attouch et al. [3]). Adam normalizes the average $\boldsymbol{m}_n$ by the second moments $\boldsymbol{v}_n$ of of the gradient $\boldsymbol{g}_n$: $\boldsymbol{v}_n = \mathrm{E}\left[\boldsymbol{g}_n \odot \boldsymbol{g}_n\right]$. $\boldsymbol{m}_n$ is componentwise divided by the square root of the components of $\boldsymbol{v}_n$. We assume that the second moments of $\boldsymbol{g}_n$ are stationary, i.e., $\boldsymbol{v} = \mathrm{E}\left[\boldsymbol{g}_n \odot \boldsymbol{g}_n\right]$. In this case the normalization can be considered as additional noise since the normalization factor randomly deviates from its mean. In the HBF interpretation the normalization by $\sqrt{\boldsymbol{v}}$ corresponds to introducing gravitation. We obtain

$$\boldsymbol{v}_n = \frac{1-\beta_2}{1-\beta_2^n} \sum_{l=1}^{n} \beta_2^{n-l} \, \boldsymbol{g}_l \odot \boldsymbol{g}_l \, , \quad \Delta\boldsymbol{v}_n = \boldsymbol{v}_n - \boldsymbol{v} = \frac{1-\beta_2}{1-\beta_2^n} \sum_{l=1}^{n} \beta_2^{n-l} \, (\boldsymbol{g}_l \odot \boldsymbol{g}_l - \boldsymbol{v}) \, . \quad (5)$$

For a stationary second moment $\boldsymbol{v}$ and $\beta_2 = 1 - \alpha a(n+1)r(n)$, we have $\Delta\boldsymbol{v}_n \propto a(n+1)r(n)$. We use a componentwise linear approximation to Adam's second moment normalization $1/\sqrt{\boldsymbol{v} + \Delta\boldsymbol{v}_n} \approx 1/\sqrt{\boldsymbol{v}} - (1/(2\boldsymbol{v} \odot \sqrt{\boldsymbol{v}})) \odot \Delta\boldsymbol{v}_n + \mathrm{O}(\Delta^2\boldsymbol{v}_n)$, where all operations are meant componentwise. If we set $\boldsymbol{M}_{n+1}^{(v)} = -(\boldsymbol{m}_n \odot \Delta\boldsymbol{v}_n)/(2\boldsymbol{v} \odot \sqrt{\boldsymbol{v}}a(n+1)r(n))$, then $\boldsymbol{m}_n/\sqrt{\boldsymbol{v}_n} \approx \boldsymbol{m}_n/\sqrt{\boldsymbol{v}} + a(n+1)r(n)\boldsymbol{M}_{n+1}^{(v)}$ and $\mathrm{E}\left[\boldsymbol{M}_{n+1}^{(v)}\right] = \boldsymbol{0}$, since $\mathrm{E}\left[\boldsymbol{g}_l \odot \boldsymbol{g}_l - \boldsymbol{v}\right] = \boldsymbol{0}$. For a stationary second moment $\boldsymbol{v}$, the random variable $\{\boldsymbol{M}_n^{(v)}\}$ is a martingale difference sequence with a bounded second moment. Therefore $\{\boldsymbol{M}_{n+1}^{(v)}\}$ can be subsumed into $\{\boldsymbol{M}_{n+1}\}$ in update rules Eq. (4). The factor $1/\sqrt{\boldsymbol{v}}$ can be componentwise incorporated into the gradient $\boldsymbol{g}$ which corresponds to rescaling the parameters without changing the minimum. □

According to Attouch et al. [3] the energy, that is, a Lyapunov function, is $E(t) = 1/2|\dot{\boldsymbol{\theta}}(t)|^2 + f(\boldsymbol{\theta}(t))$ and $\dot{E}(t) = -a\,|\dot{\boldsymbol{\theta}}(t)|^2 < 0$. Since Adam can be expressed as differential equation and has a Lyapunov function, the idea of $(T, \delta)$ perturbed ODEs [7, 22, 8] carries over to Adam. Therefore the convergence of Adam with TTUR can be proved via two time-scale stochastic approximation analysis like in Borkar [7] for stationary second moments of the gradient.

In the supplement we further discuss the convergence of two time-scale stochastic approximation algorithms with additive noise, linear update functions depending on Markov chains, nonlinear update functions, and updates depending on controlled Markov processes. Futhermore, the supplement presents work on the rate of convergence for both linear and nonlinear update rules using similar techniques as the local stability analysis of Nagarajan and Kolter [39]. Finally, we elaborate more on equal time-scale updates, which are investigated for saddle point problems and actor-critic learning.

## 3 Experiments

**Performance Measure.** Before presenting the experiments, we introduce a quality measure for models learned by GANs. The objective of generative learning is that the model produces data which matches the observed data. Therefore, each distance between the probability of observing real world data $p_w(.)$ and the probability of generating model data $p(.)$ can serve as performance measure for generative models. However, defining appropriate performance measures for generative models is difficult [45]. The best known measure is the likelihood, which can be estimated by annealed importance sampling [49]. However, the likelihood heavily depends on the noise assumptions for the real data and can be dominated by single samples [45]. Other approaches like density estimates have drawbacks, too [45]. A well-performing approach to measure the performance of GANs is the "Inception Score" which correlates with human judgment [44]. Generated samples are fed into an inception model that was trained on ImageNet. Images with meaningful objects are supposed to have low label (output) entropy, that is, they belong to few object classes. On the other hand, the entropy across images should be high, that is, the variance over the images should be large. Drawback of the Inception Score is that the statistics of real world samples are not used and compared to the statistics of synthetic samples. Next, we improve the Inception Score. The equality $p(.) = p_w(.)$ holds except for a non-measurable set if and only if $\int p(.)f(x)dx = \int p_w(.)f(x)dx$ for a basis $f(.)$ spanning the function space in which $p(.)$ and $p_w(.)$ live. These equalities of expectations

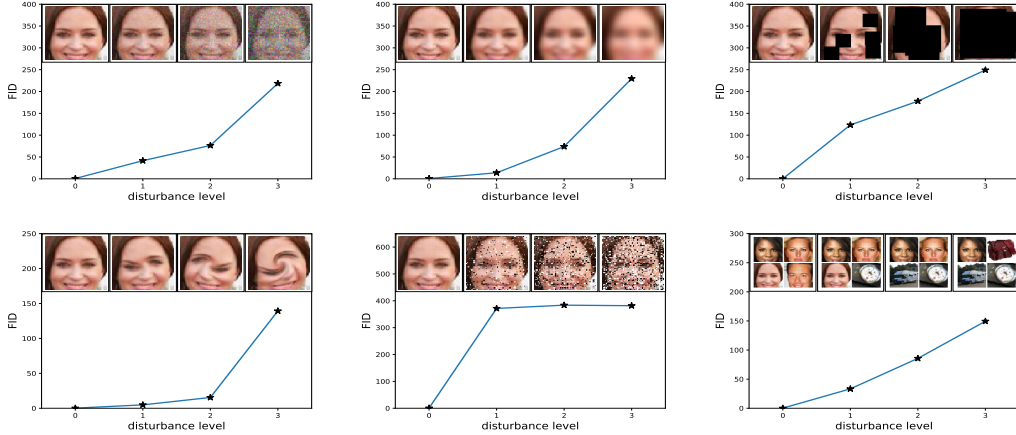

Figure 3: FID is evaluated for **upper left:** Gaussian noise, **upper middle:** Gaussian blur, **upper right:** implanted black rectangles, **lower left:** swirled images, **lower middle:** salt and pepper noise, and **lower right:** CelebA dataset contaminated by ImageNet images. The disturbance level rises from zero and increases to the highest level. The FID captures the disturbance level very well by monotonically increasing.

are used to describe distributions by moments or cumulants, where $f(x)$ are polynomials of the data $x$. We generalize these polynomials by replacing $x$ by the coding layer of an inception model in order to obtain vision-relevant features. For practical reasons we only consider the first two polynomials, that is, the first two moments: mean and covariance. The Gaussian is the maximum entropy distribution for given mean and covariance, therefore we assume the coding units to follow a multidimensional Gaussian. The difference of two Gaussians (synthetic and real-world images) is measured by the Fréchet distance [14] also known as Wasserstein-2 distance [48]. We call the Fréchet distance $d(.,.)$ between the Gaussian with mean $(\boldsymbol{m}, \boldsymbol{C})$ obtained from $p(.)$ and the Gaussian with mean $(\boldsymbol{m}_w, \boldsymbol{C}_w)$ obtained from $p_w(.)$ the "Fréchet Inception Distance" (FID), which is given by [13]: $d^2((\boldsymbol{m}, \boldsymbol{C}), (\boldsymbol{m}_w, \boldsymbol{C}_w)) = \|\boldsymbol{m} - \boldsymbol{m}_w\|_2^2 + \mathrm{Tr}\big(\boldsymbol{C} + \boldsymbol{C}_w - 2\big(\boldsymbol{C}\boldsymbol{C}_w\big)^{1/2}\big)$. Next we show that the FID is consistent with increasing disturbances and human judgment. Fig. 3 evaluates the FID for Gaussian noise, Gaussian blur, implanted black rectangles, swirled images, salt and pepper noise, and CelebA dataset contaminated by ImageNet images. The FID captures the disturbance level very well. In the experiments we used the FID to evaluate the performance of GANs. For more details and a comparison between FID and Inception Score see Supplement Section 1, where we show that FID is *more consistent* with the noise level than the Inception Score.

**Model Selection and Evaluation.** We compare the two time-scale update rule (TTUR) for GANs with the original GAN training to see whether TTUR improves the convergence speed and performance of GANs. We have selected Adam stochastic optimization to reduce the risk of mode collapsing. The advantage of Adam has been confirmed by MNIST experiments, where Adam indeed considerably reduced the cases for which we observed mode collapsing. Although TTUR ensures that the discriminator converges during learning, practicable learning rates must be found for each experiment. We face a trade-off since the learning rates should be small enough (e.g. for the generator) to ensure convergence but at the same time should be large enough to allow fast learning. For each of the experiments, the learning rates have been optimized to be large while still ensuring stable training which is indicated by a decreasing FID or Jensen-Shannon-divergence (JSD). We further fixed the time point for stopping training to the update step when the FID or Jensen-Shannon-divergence of the best models was no longer decreasing. For some models, we observed that the FID diverges or starts to increase at a certain time point. An example of this behaviour is shown in Fig. 5. The performance of generative models is evaluated via the Fréchet Inception Distance (FID) introduced above. For the One Billion Word experiment, the normalized JSD served as performance measure. For computing the FID, we propagated all images from the training dataset through the pretrained Inception-v3 model following the computation of the Inception Score [44], however, we use the last

pooling layer as coding layer. For this coding layer, we calculated the mean $\boldsymbol{m}_w$ and the covariance matrix $\boldsymbol{C}_w$. Thus, we approximate the first and second central moment of the function given by the Inception coding layer under the real world distribution. To approximate these moments for the model distribution, we generate 50,000 images, propagate them through the Inception-v3 model, and then compute the mean $\boldsymbol{m}$ and the covariance matrix $\boldsymbol{C}$. For computational efficiency, we evaluate the FID every 1,000 DCGAN mini-batch updates, every 5,000 WGAN-GP outer iterations for the image experiments, and every 100 outer iterations for the WGAN-GP language model. For the one time-scale updates a WGAN-GP outer iteration for the image model consists of five discriminator mini-batches and ten discriminator mini-batches for the language model, where we follow the original implementation. For TTUR however, the discriminator is updated only once per iteration. We repeat the training for each single time-scale (orig) and TTUR learning rate eight times for the image datasets and ten times for the language benchmark. Additionally to the mean FID training progress we show the minimum and maximum FID over all runs at each evaluation time-step. For more details, implementations and further results see Supplement Section 4 and 6.

**Simple Toy Data.** We first want to demonstrate the difference between a single time-scale update rule and TTUR on a simple toy min/max problem where a saddle point should be found. The objective $f(x,y) = (1+x^2)(100-y^2)$ in Fig. 4 (left) has a saddle point at $(x,y) = (0,0)$ and fulfills assumption A4. The norm $\|(x,y)\|$ measures the distance of the parameter vector $(x,y)$ to the saddle point. We update $(x,y)$ by gradient descent in $x$ and gradient ascent in $y$ using additive Gaussian noise in order to simulate a stochastic update. The updates should converge to the saddle point $(x,y) = (0,0)$ with objective value $f(0,0) = 100$ and the norm 0. In Fig. 4 (right), the first two rows show one time-scale update rules. The large learning rate in the first row diverges and has large fluctuations. The smaller learning rate in the second row converges but slower than the TTUR in the third row which has slow $x$-updates. TTUR with slow $y$-updates in the fourth row also converges but slower.

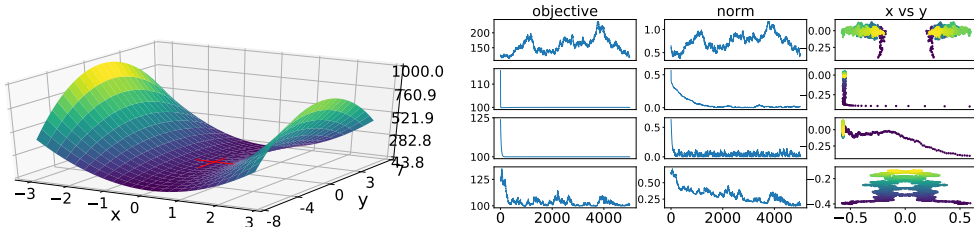

Figure 4: **Left:** Plot of the objective with a saddle point at $(0,0)$. **Right:** Training progress with equal learning rates of 0.01 (first row) and 0.001 (second row)) for $x$ and $y$, TTUR with a learning rate of 0.0001 for $x$ vs. 0.01 for $y$ (third row) and a larger learning rate of 0.01 for $x$ vs. 0.0001 for $y$ (fourth row). The columns show the function values (left), norms (middle), and $(x,y)$ (right). TTUR (third row) clearly converges faster than with equal time-scale updates and directly moves to the saddle point as shown by the norm and in the $(x,y)$-plot.

**DCGAN on Image Data.** We test TTUR for the deep convolutional GAN (DCGAN) [42] at the CelebA, CIFAR-10, SVHN and LSUN Bedrooms dataset. Fig. 5 shows the FID during learning with the original learning method (orig) and with TTUR. The original training method is faster at the beginning, but TTUR eventually achieves better performance. DCGAN trained TTUR reaches constantly a lower FID than the original method and for CelebA and LSUN Bedrooms all one time-scale runs diverge. For DCGAN the learning rate of the generator is larger then that of the discriminator, which, however, does not contradict the TTUR theory (see the Supplement Section 5). In Table 1 we report the best FID with TTUR and one time-scale training for optimized number of updates and learning rates. TTUR constantly outperforms standard training and is more stable.

**WGAN-GP on Image Data.** We used the WGAN-GP image model [21] to test TTUR with the CIFAR-10 and LSUN Bedrooms datasets. In contrast to the original code where the discriminator is trained five times for each generator update, TTUR updates the discriminator only once, therefore we align the training progress with wall-clock time. The learning rate for the original training was optimized to be large but leads to stable learning. TTUR can use a higher learning rate for the discriminator since TTUR stabilizes learning. Fig. 6 shows the FID during learning with the original

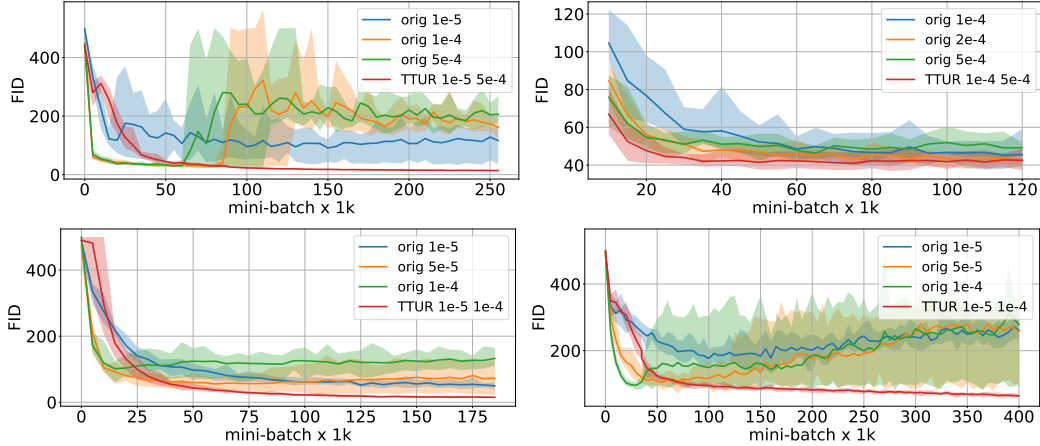

Figure 5: Mean FID (solid line) surrounded by a shaded area bounded by the maximum and the minimum over 8 runs for DCGAN on CelebA, CIFAR-10, SVHN, and LSUN Bedrooms. TTUR learning rates are given for the discriminator $b$ and generator $a$ as: "TTUR $b$ $a$". **Top Left:** CelebA. **Top Right:** CIFAR-10, starting at mini-batch update 10k for better visualisation. **Bottom Left:** SVHN. **Bottom Right:** LSUN Bedrooms. Training with TTUR (red) is more stable, has much lower variance, and leads to a better FID.

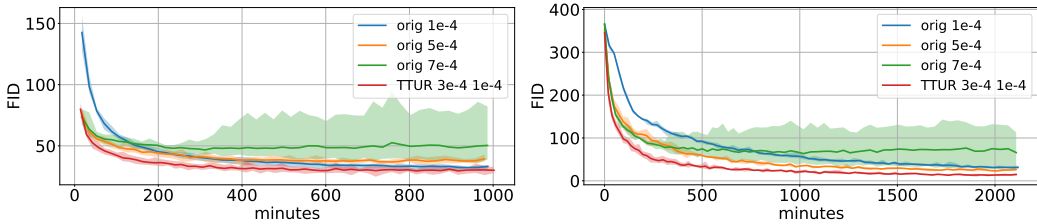

Figure 6: Mean FID (solid line) surrounded by a shaded area bounded by the maximum and the minimum over 8 runs for WGAN-GP on CelebA, CIFAR-10, SVHN, and LSUN Bedrooms. TTUR learning rates are given for the discriminator $b$ and generator $a$ as: "TTUR $b$ $a$". **Left:** CIFAR-10, starting at minute 20. **Right:** LSUN Bedrooms. Training with TTUR (red) has much lower variance and leads to a better FID.

learning method and with TTUR. Table 1 shows the best FID with TTUR and one time-scale training for optimized number of iterations and learning rates. Again TTUR reaches lower FIDs than one time-scale training.

**WGAN-GP on Language Data.** Finally the One Billion Word Benchmark [10] serves to evaluate TTUR on WGAN-GP. The character-level generative language model is a 1D convolutional neural network (CNN) which maps a latent vector to a sequence of one-hot character vectors of dimension 32 given by the maximum of a softmax output. The discriminator is also a 1D CNN applied to sequences of one-hot vectors of 32 characters. Since the FID criterium only works for images, we measured the performance by the Jensen-Shannon-divergence (JSD) between the model and the real world distribution as has been done previously [21]. In contrast to the original code where the critic is trained ten times for each generator update, TTUR updates the discriminator only once, therefore we align the training progress with wall-clock time. The learning rate for the original training was optimized to be large but leads to stable learning. TTUR can use a higher learning rate for the discriminator since TTUR stabilizes learning. We report for the 4 and 6-gram word evaluation the normalized mean JSD for ten runs for original training and TTUR training in Fig. 7. In Table 1 we report the best JSD at an optimal time-step where TTUR outperforms the standard training for both measures. The improvement of TTUR on the 6-gram statistics over original training shows that TTUR enables to learn to generate more subtle pseudo-words which better resembles real words.

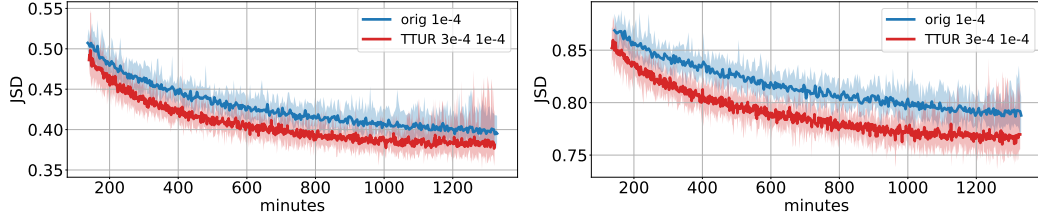

Figure 7: Performance of WGAN-GP models trained with the original (orig) and our TTUR method on the One Billion Word benchmark. The performance is measured by the normalized Jensen-Shannon-divergence based on 4-gram (**left**) and 6-gram (**right**) statistics averaged (solid line) and surrounded by a shaded area bounded by the maximum and the minimum over 10 runs, aligned to wall-clock time and starting at minute 150. TTUR learning (red) clearly outperforms the original one time-scale learning.

Table 1: The performance of DCGAN and WGAN-GP trained with the original one time-scale update rule and with TTUR on CelebA, CIFAR-10, SVHN, LSUN Bedrooms and the One Billion Word Benchmark. During training we compare the performance with respect to the FID and JSD for optimized number of updates. TTUR exhibits consistently a better FID and a better JSD.

| DCGAN Image dataset | method | b, a | updates | FID | method | b = a | updates | FID |
|---|---|---|---|---|---|---|---|---|
| CelebA | TTUR | 1e-5, 5e-4 | 225k | **12.5** | orig | 5e-4 | 70k | 21.4 |
| CIFAR-10 | TTUR | 1e-4, 5e-4 | 75k | **36.9** | orig | 1e-4 | 100k | 37.7 |
| SVHN | TTUR | 1e-5, 1e-4 | 165k | **12.5** | orig | 5e-5 | 185k | 21.4 |
| LSUN | TTUR | 1e-5, 1e-4 | 340k | **57.5** | orig | 5e-5 | 70k | 70.4 |
| WGAN-GP Image dataset | method | b, a | time(m) | FID | method | b = a | time(m) | FID |
| CIFAR-10 | TTUR | 3e-4, 1e-4 | 700 | **24.8** | orig | 1e-4 | 800 | 29.3 |
| LSUN | TTUR | 3e-4, 1e-4 | 1900 | **9.5** | orig | 1e-4 | 2010 | 20.5 |
| WGAN-GP Language n-gram | method | b, a | time(m) | JSD | method | b = a | time(m) | JSD |
| 4-gram | TTUR | 3e-4, 1e-4 | 1150 | **0.35** | orig | 1e-4 | 1040 | 0.38 |
| 6-gram | TTUR | 3e-4, 1e-4 | 1120 | **0.74** | orig | 1e-4 | 1070 | 0.77 |

## 4   Conclusion

For learning GANs, we have introduced the two time-scale update rule (TTUR), which we have proved to converge to a stationary local Nash equilibrium. Then we described Adam stochastic optimization as a heavy ball with friction (HBF) dynamics, which shows that Adam converges and that Adam tends to find flat minima while avoiding small local minima. A second order differential equation describes the learning dynamics of Adam as an HBF system. Via this differential equation, the convergence of GANs trained with TTUR to a stationary local Nash equilibrium can be extended to Adam. Finally, to evaluate GANs, we introduced the 'Fréchet Inception Distance" (FID) which captures the similarity of generated images to real ones better than the Inception Score. In experiments we have compared GANs trained with TTUR to conventional GAN training with a one time-scale update rule on CelebA, CIFAR-10, SVHN, LSUN Bedrooms, and the One Billion Word Benchmark. TTUR outperforms conventional GAN training consistently in all experiments.

## Acknowledgment

This work was supported by NVIDIA Corporation, Bayer AG with Research Agreement 09/2017, Zalando SE with Research Agreement 01/2016, Audi.JKU Deep Learning Center, Audi Electronic Venture GmbH, IWT research grant IWT150865 (Exaptation), H2020 project grant 671555 (ExCAPE) and FWF grant P 28660-N31.

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
