[Supplementary Material]

# GANs Trained by a Two Time-Scale Update Rule Converge to a Local Nash Equilibrium – Supplementary Material

**Martin Heusel**    **Hubert Ramsauer**    **Thomas Unterthiner**    **Bernhard Nessler**

**Sepp Hochreiter**

LIT AI Lab & Institute of Bioinformatics,
Johannes Kepler University Linz
A-4040 Linz, Austria
{mhe,ramsauer,unterthiner,nessler,hochreit}@bioinf.jku.at

## Abstract

We present supplementary material like background and details to the convergence proofs, analysis of the FID, additional experiments, additional figures for the paper "GANs Trained by a Two Time-Scale Update Rule Converge to a Local Nash Equilibrium"

## Contents

# 1 Fréchet Inception Distance (FID)

We improve the Inception score for comparing the results of GANs [24]. The Inception score has the disadvantage that it does not use the statistics of real world samples and compare it to the statistics of synthetic samples. Let $p(.)$ be the distribution of model samples and $p_w(.)$ the distribution of the samples from real world. The equality $p(.) = p_w(.)$ holds except for a non-measurable set if and only if $\int p(.)f(x)dx = \int p_w(.)f(x)dx$ for a basis $f(.)$ spanning the function space in which $p(.)$ and $p_w(.)$ live. These equalities of expectations are used to describe distributions by moments or cumulants, where $f(x)$ are polynomials of the data $x$. We replacing $x$ by the coding layer of an Inception model in order to obtain vision-relevant features and consider polynomials of the coding unit functions. For practical reasons we only consider the first two polynomials, that is, the first two moments: mean and covariance. The Gaussian is the maximum entropy distribution for given mean and covariance, therefore we assume the coding units to follow a multidimensional Gaussian. The difference of two Gaussians is measured by the Fréchet distance [9] also known as Wasserstein-2 distance [26]. The Fréchet distance $d(.,.)$ between the Gaussian with mean and covariance $(\boldsymbol{m}, \boldsymbol{C})$ obtained from $p(.)$ and the Gaussian $(\boldsymbol{m}_w, \boldsymbol{C}_w)$ obtained from $p_w(.)$ is called the "Fréchet Inception Distance" (FID), which is given by [8]:

$$d^2((\boldsymbol{m}, \boldsymbol{C}), (\boldsymbol{m}_w, \boldsymbol{C}_w)) = \|\boldsymbol{m} - \boldsymbol{m}_w\|_2^2 + \mathrm{Tr}\big(\boldsymbol{C} + \boldsymbol{C}_w - 2\big(\boldsymbol{C}\boldsymbol{C}_w\big)^{1/2}\big). \qquad (1)$$

Next we show that the FID is consistent with increasing disturbances and human judgment on the CelebA dataset. We computed the $(\boldsymbol{m}_w, \boldsymbol{C}_w)$ on all CelebA images, while for computing $(\boldsymbol{m}, \boldsymbol{C})$ we used 50,000 randomly selected samples. We considered following disturbances of the image $\boldsymbol{X}$:

1. **Gaussian noise**: We constructed a matrix $\boldsymbol{N}$ with Gaussian noise scaled to $[0, 255]$. The noisy image is computed as $(1 - \alpha)\boldsymbol{X} + \alpha\boldsymbol{N}$ for $\alpha \in \{0, 0.25, 0.5, 0.75\}$. The larger $\alpha$ is, the larger is the noise added to the image, the larger is the disturbance of the image.

2. **Gaussian blur**: The image is convolved with a Gaussian kernel with standard deviation $\alpha \in \{0, 1, 2, 4\}$. The larger $\alpha$ is, the larger is the disturbance of the image, that is, the more the image is smoothed.

3. **Black rectangles**: To an image five black rectangles are are added at randomly chosen locations. The rectangles cover parts of the image. The size of the rectangles is $\alpha$imagesize with $\alpha \in \{0, 0.25, 0.5, 0.75\}$. The larger $\alpha$ is, the larger is the disturbance of the image, that is, the more of the image is covered by black rectangles.

4. **Swirl**: Parts of the image are transformed as a spiral, that is, as a swirl (whirlpool effect). Consider the coordinate $(x, y)$ in the noisy (swirled) image for which we want to find the color. Towards this end we need the reverse mapping for the swirl transformation which gives the location which is mapped to $(x, y)$. We first compute polar coordinates relative to a center $(x_0, y_0)$ given by the angle $\theta = \arctan((y - y_0)/(x - x_0))$ and the radius $r = \sqrt{(x - x_0)^2 + (y - y_0)^2}$. We transform them according to $\theta' = \theta + \alpha e^{-5r/(\ln 2\rho)}$. Here $\alpha$ is a parameter for the amount of swirl and $\rho$ indicates the swirl extent in pixels. The original coordinates, where the color for $(x, y)$ can be found, are $x_{\mathrm{org}} = x_0 + r\cos(\theta')$ and $y_{\mathrm{org}} = y_0 + r\sin(\theta')$. We set $(x_0, y_0)$ to the center of the image and $\rho = 25$. The disturbance level is given by the amount of swirl $\alpha \in \{0, 1, 2, 4\}$. The larger $\alpha$ is, the larger is the disturbance of the image via the amount of swirl.

5. **Salt and pepper noise**: Some pixels of the image are set to black or white, where black is chosen with 50% probability (same for white). Pixels are randomly chosen for being flipped to white or black, where the ratio of pixel flipped to white or black is given by the noise level $\alpha \in \{0, 0.1, 0.2, 0.3\}$. The larger $\alpha$ is, the larger is the noise added to the image via flipping pixels to white or black, the larger is the disturbance level.

6. **ImageNet contamination**: From each of the 1,000 ImageNet classes, 5 images are randomly chosen, which gives 5,000 ImageNet images. The images are ensured to be RGB and to have a minimal size of 256x256. A percentage of $\alpha \in \{0, 0.25, 0.5, 0.75\}$ of the CelebA images has been replaced by ImageNet images. $\alpha = 0$ means all images are from CelebA, $\alpha = 0.25$ means that 75% of the images are from CelebA and 25% from ImageNet etc. The larger $\alpha$ is, the larger is the disturbance of the CelebA dataset by contaminating it by ImageNet images. The larger the disturbance level is, the more the dataset deviates from the reference real world dataset.

We compare the Inception Score [24] with the FID. The Inception Score with $m$ samples and $K$ classes is

$$\exp\big(\frac{1}{m}\sum_{i=1}^{m}\sum_{k=1}^{K}p(y_k\mid\boldsymbol{X}_i)\log\frac{p(y_k\mid\boldsymbol{X}_i)}{p(y_k)}\big)\,. \tag{2}$$

The FID is a distance, while the Inception Score is a score. To compare FID and Inception Score, we transform the Inception Score to a distance, which we call "Inception Distance" (IND). This transformation to a distance is possible since the Inception Score has a maximal value. For zero probability $p(y_k\mid\boldsymbol{X}_i)=0$, we set the value $p(y_k\mid\boldsymbol{X}_i)\log\frac{p(y_k\mid\boldsymbol{X}_i)}{p(y_k)}=0$. We can bound the log-term by

$$\log\frac{p(y_k\mid\boldsymbol{X}_i)}{p(y_k)}\;\leqslant\;\log\frac{1}{1/m}\;=\;\log m\,. \tag{3}$$

Using this bound, we obtain an upper bound on the Inception Score:

$$\exp\big(\frac{1}{m}\sum_{i=1}^{m}\sum_{k=1}^{K}p(y_k\mid\boldsymbol{X}_i)\log\frac{p(y_k\mid\boldsymbol{X}_i)}{p(y_k)}\big) \tag{4}$$

$$\leqslant\;\exp\big(\log m\frac{1}{m}\sum_{i=1}^{m}\sum_{k=1}^{K}p(y_k\mid\boldsymbol{X}_i)\big) \tag{5}$$

$$=\;\exp\big(\log m\frac{1}{m}\sum_{i=1}^{m}1\big)\;=\;m\,. \tag{6}$$

The upper bound is tight and achieved if $m\leqslant K$ and every sample is from a different class and the sample is classified correctly with probability 1. The IND is computed "IND = $m$ - Inception Score", therefore the IND is zero for a perfect subset of the ImageNet with $m<K$ samples, where each sample stems from a different class. Therefore both distances should increase with increasing disturbance level. In Figure 1 we present the evaluation for each kind of disturbance. The larger the disturbance level is, the larger the FID and IND should be. In Figure 2, 3, 4, and 4 we show examples of images generated with DCGAN trained on CelebA with FIDs 500, 300, 133, 100, 45, 13, and FID 3 achieved with WGAN-GP on CelebA.

Figure 1: **Left:** FID and **right:** Inception Score are evaluated for **first row:** Gaussian noise, **second row:** Gaussian blur, **third row:** implanted black rectangles, **fourth row:** swirled images, **fifth row.** salt and pepper noise, and **sixth row:** the CelebA dataset contaminated by ImageNet images. Left is the smallest disturbance level of zero, which increases to the highest level at right. The FID captures the disturbance level very well by monotonically increasing whereas the Inception Score fluctuates, stays flat or even, in the worst case, decreases.

Figure 2: Samples generated from DCGAN trained on CelebA with different FIDs. **Left:** FID 500 and **Right:** FID 300.

Figure 3: Samples generated from DCGAN trained on CelebA with different FIDs. **Left:** FID 133 and **Right:** FID 100.

Figure 4: Samples generated from DCGAN trained on CelebA with different FIDs. **Left:** FID 45 and **Right:** FID 13.

Figure 5: Samples generated from WGAN-GP trained on CelebA with a FID of 3.

## 2 Two Time-Scale Stochastic Approximation Algorithms

Stochastic approximation algorithms are iterative procedures to find a root or a stationary point (minimum, maximum, saddle point) of a function when only noisy observations of its values or its derivatives are provided. Two time-scale stochastic approximation algorithms are two coupled iterations with different step sizes. For proving convergence of these interwoven iterates it is assumed that one step size is considerably smaller than the other. The slower iterate (the one with smaller step size) is assumed to be slow enough to allow the fast iterate converge while being perturbed by the the slower. The perturbations of the slow should be small enough to ensure convergence of the faster.

The iterates map at time step $n \geqslant 0$ the fast variable $\boldsymbol{w}_n \in \mathbb{R}^k$ and the slow variable $\boldsymbol{\theta}_n \in \mathbb{R}^m$ to their new values:

$$\boldsymbol{\theta}_{n+1} \;=\; \boldsymbol{\theta}_n \;+\; a(n) \; \left( \boldsymbol{h}\big(\boldsymbol{\theta}_n, \boldsymbol{w}_n, \boldsymbol{Z}_n^{(\theta)}\big) \;+\; \boldsymbol{M}_n^{(\theta)} \right) \;, \tag{7}$$

$$\boldsymbol{w}_{n+1} \;=\; \boldsymbol{w}_n \;+\; b(n) \; \left( \boldsymbol{g}\big(\boldsymbol{\theta}_n, \boldsymbol{w}_n, \boldsymbol{Z}_n^{(w)}\big) \;+\; \boldsymbol{M}_n^{(w)} \right) \;. \tag{8}$$

The iterates use

- $\boldsymbol{h}(.) \in \mathbb{R}^m$: mapping for the slow iterate Eq. (7),
- $\boldsymbol{g}(.) \in \mathbb{R}^k$: mapping for the fast iterate Eq. (8),
- $a(n)$: step size for the slow iterate Eq. (7),
- $b(n)$: step size for the fast iterate Eq. (8),
- $\boldsymbol{M}_n^{(\theta)}$: additive random Markov process for the slow iterate Eq. (7),
- $\boldsymbol{M}_n^{(w)}$: additive random Markov process for the fast iterate Eq. (8),
- $\boldsymbol{Z}_n^{(\theta)}$: random Markov process for the slow iterate Eq. (7),
- $\boldsymbol{Z}_n^{(w)}$: random Markov process for the fast iterate Eq. (8).

### 2.1 Convergence of Two Time-Scale Stochastic Approximation Algorithms

#### 2.1.1 Additive Noise

The first result is from Borkar 1997 [5] which was generalized in Konda and Borkar 1999 [15]. Borkar considered the iterates:

$$\boldsymbol{\theta}_{n+1} \;=\; \boldsymbol{\theta}_n \;+\; a(n) \; \left( \boldsymbol{h}\big(\boldsymbol{\theta}_n, \boldsymbol{w}_n\big) \;+\; \boldsymbol{M}_n^{(\theta)} \right) \;, \tag{9}$$

$$\boldsymbol{w}_{n+1} \;=\; \boldsymbol{w}_n \;+\; b(n) \; \left( \boldsymbol{g}\big(\boldsymbol{\theta}_n, \boldsymbol{w}_n\big) \;+\; \boldsymbol{M}_n^{(w)} \right) \;. \tag{10}$$

**Assumptions.**  We make the following assumptions:

(A1) Assumptions on the update functions: The functions $\boldsymbol{h} : \mathbb{R}^{k+m} \mapsto \mathbb{R}^m$ and $\boldsymbol{g} : \mathbb{R}^{k+m} \mapsto \mathbb{R}^k$ are Lipschitz.

(A2) Assumptions on the learning rates:

$$\sum_n a(n) \;=\; \infty \quad , \quad \sum_n a^2(n) \;<\; \infty \;, \tag{11}$$

$$\sum_n b(n) \;=\; \infty \quad , \quad \sum_n b^2(n) \;<\; \infty \;, \tag{12}$$

$$a(n) \;=\; \mathrm{o}(b(n)) \;, \tag{13}$$

(A3) Assumptions on the noise: For the increasing $\sigma$-field

$$\mathcal{F}_n \;=\; \sigma(\boldsymbol{\theta}_l, \boldsymbol{w}_l, \boldsymbol{M}_l^{(\theta)}, \boldsymbol{M}_l^{(w)}, l \leqslant n), n \geqslant 0 \;,$$

the sequences of random variables $(\boldsymbol{M}_n^{(\theta)}, \mathcal{F}_n)$ and $(\boldsymbol{M}_n^{(w)}, \mathcal{F}_n)$ satisfy

$$\sum_n a(n)\, \boldsymbol{M}_n^{(\theta)} \; < \; \infty \text{ a.s.} \tag{14}$$

$$\sum_n b(n)\, \boldsymbol{M}_n^{(w)} \; < \; \infty \text{ a.s. .} \tag{15}$$

**(A4)** Assumption on the existence of a solution of the fast iterate: For each $\boldsymbol{\theta} \in \mathbb{R}^m$, the ODE

$$\dot{\boldsymbol{w}}(t) \; = \; \boldsymbol{g}\big(\boldsymbol{\theta}, \boldsymbol{w}(t)\big) \tag{16}$$

has a unique global asymptotically stable equilibrium $\boldsymbol{\lambda}(\boldsymbol{\theta})$ such that $\boldsymbol{\lambda} : \mathbb{R}^m \mapsto \mathbb{R}^k$ is Lipschitz.

**(A5)** Assumption on the existence of a solution of the slow iterate: The ODE

$$\dot{\boldsymbol{\theta}}(t) \; = \; \boldsymbol{h}\big(\boldsymbol{\theta}(t), \boldsymbol{\lambda}(\boldsymbol{\theta}(t))\big) \tag{17}$$

has a unique global asymptotically stable equilibrium $\boldsymbol{\theta}^*$.

**(A6)** Assumption of bounded iterates:

$$\sup_n \|\boldsymbol{\theta}_n\| \; < \; \infty \,, \tag{18}$$

$$\sup_n \|\boldsymbol{w}_n\| \; < \; \infty \,. \tag{19}$$

**Convergence Theorem**  The next theorem is from Borkar 1997 [5].

**Theorem 1** (Borkar). *If the assumptions are satisfied, then the iterates Eq. (9) and Eq. (10) converge to $(\boldsymbol{\theta}^*, \boldsymbol{\lambda}(\boldsymbol{\theta}^*))$ a.s.*

**Comments**

**(C1)** According to Lemma 2 in [4] Assumption (A3) is fulfilled if $\{\boldsymbol{M}_n^{(\theta)}\}$ is a martingale difference sequence w.r.t $\mathcal{F}_n$ with

$$\mathrm{E}\left[\|\boldsymbol{M}_n^{(\theta)}\|^2 \mid \mathcal{F}_n^{(\theta)}\right] \; \leqslant \; B_1$$

and $\{\boldsymbol{M}_n^{(w)}\}$ is a martingale difference sequence w.r.t $\mathcal{F}_n$ with

$$\mathrm{E}\left[\|\boldsymbol{M}_n^{(w)}\|^2 \mid \mathcal{F}_n^{(w)}\right] \; \leqslant \; B_2 \,,$$

where $B_1$ and $B_2$ are positive deterministic constants.

**(C2)** Assumption (A3) holds for mini-batch learning which is the most frequent case of stochastic gradient. The batch gradient is $\boldsymbol{G}_n := \nabla_\theta\big(\frac{1}{N}\sum_{i=1}^N f(\boldsymbol{x}_i, \theta)\big), 1 \leqslant i \leqslant N$ and the mini-batch gradient for batch size $s$ is $\boldsymbol{h}_n := \nabla_\theta\big(\frac{1}{s}\sum_{i=1}^s f(\boldsymbol{x}_{u_i}, \theta)\big), 1 \leqslant u_i \leqslant N$, where the indexes $u_i$ are randomly and uniformly chosen. For the noise $\boldsymbol{M}_n^{(\theta)} := \boldsymbol{h}_n - \boldsymbol{G}_n$ we have $\mathrm{E}[\boldsymbol{M}_n^{(\theta)}] = \mathrm{E}[\boldsymbol{h}_n] - \boldsymbol{G}_n = \boldsymbol{G}_n - \boldsymbol{G}_n = 0$. Since the indexes are chosen without knowing past events, we have a martingale difference sequence. For bounded gradients we have bounded $\|\boldsymbol{M}_n^{(\theta)}\|^2$.

**(C3)** We address assumption (A4) with weight decay in two ways: (I) Weight decay avoids problems with a discriminator that is region-wise constant and, therefore, does not have a locally stable generator. If the generator is perfect, then the discriminator is 0.5 everywhere. For generator with mode collapse, (i) the discriminator is 1 in regions without generator examples, (ii) 0 in regions with generator examples only, (iii) is equal to the local ratio of real world examples for regions with generator and real world examples. Since the discriminator is locally constant, the generator has gradient zero and cannot improve. Also the discriminator cannot improve, since it has minimal error given the current generator. However, without weight decay the Nash Equilibrium is not stable since the second order derivatives are zero, too. (II) Weight decay avoids that the generator is driven to infinity with unbounded weights. For example a linear discriminator can supply a gradient for the generator outside each bounded region.

**(C4)** The main result used in the proof of the theorem relies on work on perturbations of ODEs according to Hirsch 1989 [11].

**(C5)** Konda and Borkar 1999 [15] generalized the convergence proof to distributed asynchronous update rules.

**(C6)** Tadić relaxed the assumptions for showing convergence [25]. In particular the noise assumptions (Assumptions A2 in [25]) do not have to be martingale difference sequences and are more general than in [5]. In another result the assumption of bounded iterates is not necessary if other assumptions are ensured [25]. Finally, Tadić considers the case of non-additive noise [25]. **Tadić does not provide proofs for his results.** We were not able to find such proofs even in other publications of Tadić.

### 2.1.2 Linear Update, Additive Noise, and Markov Chain

In contrast to the previous subsection, we assume that an additional Markov chain influences the iterates [14, 16]. The Markov chain allows applications in reinforcement learning, in particular in actor-critic setting where the Markov chain is used to model the environment. The slow iterate is the actor update while the fast iterate is the critic update. For reinforcement learning both the actor and the critic observe the environment which is driven by the actor actions. The environment observations are assumed to be a Markov chain. The Markov chain can include eligibility traces which are modeled as explicit states in order to keep the Markov assumption.

The Markov chain is the sequence of observations of the environment which progresses via transition probabilities. The transitions are not affected by the critic but by the actor.

Konda et al. considered the iterates [14, 16]:

$$\boldsymbol{\theta}_{n+1} \;=\; \boldsymbol{\theta}_n \;+\; a(n)\,\boldsymbol{H}_n \;, \tag{20}$$

$$\boldsymbol{w}_{n+1} \;=\; \boldsymbol{w}_n \;+\; b(n)\,\left(\boldsymbol{g}\big(\boldsymbol{Z}_n^{(w)};\boldsymbol{\theta}_n\big) \;+\; \boldsymbol{G}\big(\boldsymbol{Z}_n^{(w)};\boldsymbol{\theta}_n\big)\,\boldsymbol{w}_n \;+\; \boldsymbol{M}_n^{(w)}\,\boldsymbol{w}_n\right) \;. \tag{21}$$

$\boldsymbol{H}_n$ is a random process that drives the changes of $\boldsymbol{\theta}_n$. We assume that $\boldsymbol{H}_n$ is a slow enough process. We have a linear update rule for the fast iterate using the vector function $\boldsymbol{g}(.) \in \mathbb{R}^k$ and the matrix function $\boldsymbol{G}(.) \in \mathbb{R}^{k \times k}$.

**Assumptions.** We make the following assumptions:

**(A1)** Assumptions on the Markov process, that is, the transition kernel: The stochastic process $\boldsymbol{Z}_n^{(w)}$ takes values in a Polish (complete, separable, metric) space $\mathbb{Z}$ with the Borel $\sigma$-field

$$\mathcal{F}_n \;=\; \sigma(\boldsymbol{\theta}_l, \boldsymbol{w}_l, \boldsymbol{Z}_l^{(w)}, \boldsymbol{H}_l, l \leqslant n), n \geqslant 0 \;.$$

For every measurable set $A \subset \mathbb{Z}$ and the parametrized transition kernel $\mathrm{P}(.;\boldsymbol{\theta}_n)$ we have:

$$\mathrm{P}(\boldsymbol{Z}_{n+1}^{(w)} \in A \mid \mathcal{F}_n) \;=\; \mathrm{P}(\boldsymbol{Z}_{n+1}^{(w)} \in A \mid \boldsymbol{Z}_n^{(w)};\boldsymbol{\theta}_n) \;=\; \mathrm{P}(\boldsymbol{Z}_n^{(w)}, A;\boldsymbol{\theta}_n) \;. \tag{22}$$

We define for every measurable function $f$

$$\mathrm{P}_{\boldsymbol{\theta}} f(\boldsymbol{z}) \;:=\; \int \mathrm{P}(\boldsymbol{z}, \mathrm{d}\bar{\boldsymbol{z}};\boldsymbol{\theta}_n)\, f(\bar{\boldsymbol{z}}) \;.$$

**(A2)** Assumptions on the learning rates:

$$\sum_n b(n) \;=\; \infty \quad , \quad \sum_n b^2(n) \;<\; \infty \;, \tag{23}$$

$$\sum_n \left(\frac{a(n)}{b(n)}\right)^d \;<\; \infty \;, \tag{24}$$

for some $d > 0$.

**(A3)** Assumptions on the noise: The sequence $\boldsymbol{M}_n^{(w)}$ is a $k \times k$-matrix valued $\mathcal{F}_n$-martingale difference with bounded moments:

$$\mathrm{E}\left[\boldsymbol{M}_n^{(w)} \mid \mathcal{F}_n\right] \;=\; 0 \;, \tag{25}$$

$$\sup_n \mathrm{E}\left[\left\|\boldsymbol{M}_n^{(w)}\right\|^d\right] \;<\; \infty \;, \;\forall d > 0 \;. \tag{26}$$

We assume slowly changing $\boldsymbol{\theta}$, therefore the random process $\boldsymbol{H}_n$ satisfies

$$\sup_n \mathrm{E}\left[\|\boldsymbol{H}_n\|^d\right] \;<\; \infty\,,\;\forall d > 0\,. \tag{27}$$

(A4) Assumption on the existence of a solution of the fast iterate: We assume the existence of a solution to the Poisson equation for the fast iterate. For each $\boldsymbol{\theta} \in \mathbb{R}^m$, there exist functions $\bar{\boldsymbol{g}}(\boldsymbol{\theta}) \in \mathbb{R}^k$, $\bar{\boldsymbol{G}}(\boldsymbol{\theta}) \in \mathbb{R}^{k \times k}$, $\hat{\boldsymbol{g}}(\boldsymbol{z};\boldsymbol{\theta}) : \mathbb{Z} \to \mathbb{R}^k$, and $\hat{\boldsymbol{G}}(\boldsymbol{z};\boldsymbol{\theta}) : \mathbb{Z} \to \mathbb{R}^{k \times k}$ that satisfy the Poisson equations:

$$\hat{\boldsymbol{g}}(\boldsymbol{z};\boldsymbol{\theta}) \;=\; \boldsymbol{g}(\boldsymbol{z};\boldsymbol{\theta}) \;-\; \bar{\boldsymbol{g}}(\boldsymbol{\theta}) \;+\; (\mathrm{P}_{\boldsymbol{\theta}}\hat{\boldsymbol{g}}(.;\boldsymbol{\theta}))(\boldsymbol{z})\,, \tag{28}$$

$$\hat{\boldsymbol{G}}(\boldsymbol{z};\boldsymbol{\theta}) \;=\; \boldsymbol{G}(\boldsymbol{z};\boldsymbol{\theta}) \;-\; \bar{\boldsymbol{G}}(\boldsymbol{\theta}) \;+\; (\mathrm{P}_{\boldsymbol{\theta}}\hat{\boldsymbol{G}}(.;\boldsymbol{\theta}))(\boldsymbol{z})\,. \tag{29}$$

(A5) Assumptions on the update functions and solutions to the Poisson equation:

(a) Boundedness of solutions: For some constant $C$ and for all $\boldsymbol{\theta}$:

$$\max\{\|\bar{g}(\boldsymbol{\theta})\|\} \;\leqslant\; C\,, \tag{30}$$

$$\max\{\|\bar{G}(\boldsymbol{\theta})\|\} \;\leqslant\; C\,. \tag{31}$$

(b) Boundedness in expectation: All moments are bounded. For any $d > 0$, there exists $C_d > 0$ such that

$$\sup_n \mathrm{E}\left[\left\|\hat{\boldsymbol{g}}(\boldsymbol{Z}_n^{(w)};\boldsymbol{\theta})\right\|^d\right] \;\leqslant\; C_d\,, \tag{32}$$

$$\sup_n \mathrm{E}\left[\left\|\boldsymbol{g}(\boldsymbol{Z}_n^{(w)};\boldsymbol{\theta})\right\|^d\right] \;\leqslant\; C_d\,, \tag{33}$$

$$\sup_n \mathrm{E}\left[\left\|\hat{\boldsymbol{G}}(\boldsymbol{Z}_n^{(w)};\boldsymbol{\theta})\right\|^d\right] \;\leqslant\; C_d\,, \tag{34}$$

$$\sup_n \mathrm{E}\left[\left\|\boldsymbol{G}(\boldsymbol{Z}_n^{(w)};\boldsymbol{\theta})\right\|^d\right] \;\leqslant\; C_d\,. \tag{35}$$

(c) Lipschitz continuity of solutions: For some constant $C > 0$ and for all $\boldsymbol{\theta},\bar{\boldsymbol{\theta}} \in \mathbb{R}^m$:

$$\left\|\bar{\boldsymbol{g}}(\boldsymbol{\theta}) \;-\; \bar{\boldsymbol{g}}(\bar{\boldsymbol{\theta}})\right\| \;\leqslant\; C\left\|\boldsymbol{\theta} - \bar{\boldsymbol{\theta}}\right\|\,, \tag{36}$$

$$\left\|\bar{\boldsymbol{G}}(\boldsymbol{\theta}) \;-\; \bar{\boldsymbol{G}}(\bar{\boldsymbol{\theta}})\right\| \;\leqslant\; C\left\|\boldsymbol{\theta} - \bar{\boldsymbol{\theta}}\right\|\,. \tag{37}$$

(d) Lipschitz continuity in expectation: There exists a positive measurable function $C(.)$ on $\mathbb{Z}$ such that

$$\sup_n \mathrm{E}\left[C(\boldsymbol{Z}_n^{(w)})^d\right] \;<\; \infty\,,\;\forall d > 0\,. \tag{38}$$

Function $C(.)$ gives the Lipschitz constant for every $\boldsymbol{z}$:

$$\left\|(\mathrm{P}_{\boldsymbol{\theta}}\hat{\boldsymbol{g}}(.;\boldsymbol{\theta}))(\boldsymbol{z}) \;-\; (\mathrm{P}_{\bar{\boldsymbol{\theta}}}\hat{\boldsymbol{g}}(.;\bar{\boldsymbol{\theta}}))(\boldsymbol{z})\right\| \;\leqslant\; C(\boldsymbol{z})\left\|\boldsymbol{\theta} - \bar{\boldsymbol{\theta}}\right\|\,, \tag{39}$$

$$\left\|(\mathrm{P}_{\boldsymbol{\theta}}\hat{\boldsymbol{G}}(.;\boldsymbol{\theta}))(\boldsymbol{z}) \;-\; (\mathrm{P}_{\bar{\boldsymbol{\theta}}}\hat{\boldsymbol{G}}(.;\bar{\boldsymbol{\theta}}))(\boldsymbol{z})\right\| \;\leqslant\; C(\boldsymbol{z})\left\|\boldsymbol{\theta} - \bar{\boldsymbol{\theta}}\right\|\,. \tag{40}$$

(e) Uniform positive definiteness: There exists some $\alpha > 0$ such that for all $\boldsymbol{w} \in \mathbb{R}^k$ and $\boldsymbol{\theta} \in \mathbb{R}^m$:

$$\boldsymbol{w}^T\,\bar{G}(\boldsymbol{\theta})\,\boldsymbol{w} \;\geqslant\; \alpha\,\|\boldsymbol{w}\|^2\,. \tag{41}$$

**Convergence Theorem.**  We report Theorem 3.2 (see also Theorem 7 in [16]) and Theorem 3.13 from [14]:

**Theorem 2** (Konda & Tsitsiklis). *If the assumptions are satisfied, then for the iterates Eq. (20) and Eq. (21) holds:*

$$\lim_{n \to \infty}\left\|\bar{G}(\boldsymbol{\theta}_n)\,\boldsymbol{w}_n \;-\; \bar{g}(\boldsymbol{\theta}_n)\right\| \;=\; 0\;\;a.s.\,, \tag{42}$$

$$\lim_{n \to \infty}\left\|\boldsymbol{w}_n \;-\; \bar{G}^{-1}(\boldsymbol{\theta}_n)\,\bar{g}(\boldsymbol{\theta}_n)\right\| \;=\; 0\,. \tag{43}$$

**Comments.**

(C1) The proofs only use the boundedness of the moments of $H_n$ [14, 16], therefore $H_n$ may depend on $w_n$. In his PhD thesis [14], Vijaymohan Konda used this framework for the actor-critic learning, where $H_n$ drives the updates of the actor parameters $\theta_n$. However, the actor updates are based on the current parameters $w_n$ of the critic.

(C2) The random process $Z_n^{(w)}$ can affect $H_n$ as long as boundedness is ensured.

(C3) Nonlinear update rule. $g(Z_n^{(w)};\theta_n) + G(Z_n^{(w)};\theta_n)w_n$ can be viewed as a linear approximation of a nonlinear update rule. The nonlinear case has been considered in [14] where additional approximation errors due to linearization were addressed. These errors are treated in the given framework [14].

### 2.1.3 Additive Noise and Controlled Markov Processes

The most general iterates use nonlinear update functions $g$ and $h$, have additive noise, and have controlled Markov processes [12].

$$\theta_{n+1} = \theta_n + a(n)\left(h(\theta_n, w_n, Z_n^{(\theta)}) + M_n^{(\theta)}\right), \tag{44}$$

$$w_{n+1} = w_n + b(n)\left(g(\theta_n, w_n, Z_n^{(w)}) + M_n^{(w)}\right). \tag{45}$$

**Required Definitions.** *Marchaud Map*: A set-valued map $h : \mathbb{R}^l \to \{\text{subsets of } \mathbb{R}^k\}$ is called a *Marchaud map* if it satisfies the following properties:

(i) For each $\theta \in \mathbb{R}^l$, $h(\theta)$ is convex and compact.

(ii) *(point-wise boundedness)* For each $\theta \in \mathbb{R}^l$, $\sup_{w \in h(\theta)} \|w\| < K(1 + \|\theta\|)$ for some $K > 0$.

(iii) $h$ is an *upper-semicontinuous* map.
We say that $h$ is upper-semicontinuous, if given sequences $\{\theta_n\}_{n \geq 1}$ (in $\mathbb{R}^l$) and $\{y_n\}_{n \geq 1}$ (in $\mathbb{R}^k$) with $\theta_n \to \theta$, $y_n \to y$ and $y_n \in h(\theta_n), n \geq 1, y \in h(\theta)$. In other words, the graph of $h$, $\{(x, y) : y \in h(x), x \in \mathbb{R}^l\}$, is closed in $\mathbb{R}^l \times \mathbb{R}^k$.

If the set-valued map $H : \mathbb{R}^m \to \{\text{subsets of } \mathbb{R}^m\}$ is Marchaud, then the differential inclusion (DI) given by

$$\dot{\theta}(t) \in H(\theta(t)) \tag{46}$$

is guaranteed to have at least one solution that is absolutely continuous. If $\Theta$ is an absolutely continuous map satisfying Eq. (46) then we say that $\Theta \in \Sigma$.

*Invariant Set*: $M \subseteq \mathbb{R}^m$ is *invariant* if for every $\theta \in M$ there exists a trajectory, $\Theta$, entirely in $M$ with $\Theta(0) = \theta$. In other words, $\Theta \in \Sigma$ with $\Theta(t) \in M$, for all $t \geq 0$.
*Internally Chain Transitive Set*: $M \subset \mathbb{R}^m$ is said to be internally chain transitive if $M$ is compact and for every $\theta, y \in M, \epsilon > 0$ and $T > 0$ we have the following: There exist $\Phi^1, \ldots, \Phi^n$ that are $n$ solutions to the differential inclusion $\dot{\theta}(t) \in h(\theta(t))$, a sequence $\theta_1(= \theta), \ldots, \theta_{n+1}(= y) \subset M$ and $n$ real numbers $t_1, t_2, \ldots, t_n$ greater than $T$ such that: $\Phi_{t_i}^i(\theta_i) \in N^\epsilon(\theta_{i+1})$ where $N^\epsilon(\theta)$ is the open $\epsilon$-neighborhood of $\theta$ and $\Phi_{[0,t_i]}^i(\theta_i) \subset M$ for $1 \leq i \leq n$. The sequence $(\theta_1(= \theta), \ldots, \theta_{n+1}(= y))$ is called an $(\epsilon, T)$ chain in $M$ from $\theta$ to $y$.

**Assumptions.** We make the following assumptions [12]:

(A1) Assumptions on the controlled Markov processes: The controlled Markov process $\{Z_n^{(w)}\}$ takes values in a compact metric space $S^{(w)}$. The controlled Markov process $\{Z_n^{(\theta)}\}$ takes values in a compact metric space $S^{(\theta)}$. Both processes are controlled by the iterate sequences $\{\theta_n\}$ and $\{w_n\}$. Furthermore $\{Z_n^{(w)}\}$ is additionally controlled by a random process $\{A_n^{(w)}\}$ taking values in a compact metric space $U^{(w)}$ and $\{Z_n^{(\theta)}\}$ is additionally

controlled by a random process $\{A_n^{(\theta)}\}$ taking values in a compact metric space $U^{(\theta)}$. The $\{Z_n^{(\theta)}\}$ dynamics is

$$\mathrm{P}(Z_{n+1}^{(\theta)} \in B^{(\theta)} | Z_l^{(\theta)}, A_l^{(\theta)}, \theta_l, w_l, l \leqslant n) = \int_{B^{(\theta)}} p^{(\theta)}(\mathrm{d}z | Z_n^{(\theta)}, A_n^{(\theta)}, \theta_n, w_n), n \geqslant 0 ,$$
(47)

for $B^{(\theta)}$ Borel in $S^{(\theta)}$. The $\{Z_n^{(w)}\}$ dynamics is

$$\mathrm{P}(Z_{n+1}^{(w)} \in B^{(w)} | Z_l^{(w)}, A_l^{(w)}, \theta_l, w_l, l \leqslant n) = \int_{B^{(w)}} p^{(w)}(\mathrm{d}z | Z_n^{(w)}, A_n^{(w)}, \theta_n, w_n), n \geqslant 0 ,$$
(48)

for $B^{(w)}$ Borel in $S^{(w)}$.

(A2) Assumptions on the update functions: $h : \mathbb{R}^{m+k} \times S^{(\theta)} \to \mathbb{R}^m$ is jointly continuous as well as Lipschitz in its first two arguments uniformly w.r.t. the third. The latter condition means that

$$\forall z^{(\theta)} \in S^{(\theta)} : \|h(\theta, w, z^{(\theta)}) - h(\theta', w', z^{(\theta)})\| \leqslant L^{(\theta)} \left(\|\theta - \theta'\| + \|w - w'\|\right) .$$
(49)

Note that the Lipschitz constant $L^{(\theta)}$ does not depend on $z^{(\theta)}$.

$g : \mathbb{R}^{k+m} \times S^{(w)} \to \mathbb{R}^k$ is jointly continuous as well as Lipschitz in its first two arguments uniformly w.r.t. the third. The latter condition means that

$$\forall z^{(w)} \in S^{(w)} : \|g(\theta, w, z^{(w)}) - g(\theta', w', z^{(w)})\| \leqslant L^{(w)} \left(\|\theta - \theta'\| + \|w - w'\|\right) .$$
(50)

Note that the Lipschitz constant $L^{(w)}$ does not depend on $z^{(w)}$.

(A3) Assumptions on the additive noise: $\{M_n^{(\theta)}\}$ and $\{M_n^{(w)}\}$ are martingale difference sequence with second moments bounded by $K(1 + \|\theta_n\|^2 + \|w_n\|^2)$. More precisely, $\{M_n^{(\theta)}\}$ is a martingale difference sequence w.r.t. increasing $\sigma$-fields

$$\mathcal{F}_n = \sigma(\theta_l, w_l, M_l^{(\theta)}, M_l^{(w)}, Z_l^{(\theta)}, Z_l^{(w)}, l \leqslant n), \ n \geqslant 0 ,$$
(51)

satisfying

$$\mathrm{E}\left[\|M_{n+1}^{(\theta)}\|^2 \mid \mathcal{F}_n\right] \leqslant K\left(1 + \|\theta_n\|^2 + \|w_n\|^2\right) ,$$
(52)

for $n \geqslant 0$ and a given constant $K > 0$.

$\{M_n^{(w)}\}$ is a martingale difference sequence w.r.t. increasing $\sigma$-fields

$$\mathcal{F}_n = \sigma(\theta_l, w_l, M_l^{(\theta)}, M_l^{(w)}, Z_l^{(\theta)}, Z_l^{(w)}, l \leqslant n), \ n \geqslant 0 ,$$
(53)

satisfying

$$\mathrm{E}\left[\|M_{n+1}^{(w)}\|^2 \mid \mathcal{F}_n\right] \leqslant K\left(1 + \|\theta_n\|^2 + \|w_n\|^2\right) ,$$
(54)

for $n \geqslant 0$ and a given constant $K > 0$.

(A4) Assumptions on the learning rates:

$$\sum_n a(n) = \infty \quad , \quad \sum_n a^2(n) < \infty ,$$
(55)

$$\sum_n b(n) = \infty \quad , \quad \sum_n b^2(n) < \infty ,$$
(56)

$$a(n) = \mathrm{o}(b(n)) ,$$
(57)

Furthermore, $a(n), b(n), n \geqslant 0$ are non-increasing.

**(A5)** Assumptions on the controlled Markov processes, that is, the transition kernels: The state-action map

$$S^{(\theta)} \times U^{(\theta)} \times \mathbb{R}^{m+k} \ni (\boldsymbol{z}^{(\theta)}, \boldsymbol{a}^{(\theta)}, \boldsymbol{\theta}, \boldsymbol{w}) \rightarrow \mathrm{p}^{(\theta)}(\mathrm{d}\boldsymbol{y} \mid \boldsymbol{z}^{(\theta)}, \boldsymbol{a}^{(\theta)}, \boldsymbol{\theta}, \boldsymbol{w}) \quad (58)$$

and the state-action map

$$S^{(w)} \times U^{(w)} \times \mathbb{R}^{m+k} \ni (\boldsymbol{z}^{(w)}, \boldsymbol{a}^{(w)}, \boldsymbol{\theta}, \boldsymbol{w}) \rightarrow \mathrm{p}^{(w)}(\mathrm{d}\boldsymbol{y} \mid \boldsymbol{z}^{(w)}, \boldsymbol{a}^{(w)}, \boldsymbol{\theta}, \boldsymbol{w}) \quad (59)$$

are continuous.

**(A6)** Assumptions on the existence of a solution:

We consider *occupation measures* which give for the controlled Markov process the probability or density to observe a particular state-action pair from $S \times U$ for given $\boldsymbol{\theta}$ and a given control policy $\pi$. We denote by $D^{(w)}(\boldsymbol{\theta}, \boldsymbol{w})$ the set of all ergodic occupation measures for the prescribed $\boldsymbol{\theta}$ and $\boldsymbol{w}$ on state-action space $S^{(w)} \times U^{(\theta)}$ for the controlled Markov process $\boldsymbol{Z}^{(w)}$ with policy $\pi^{(w)}$. Analogously we denote, by $D^{(\theta)}(\boldsymbol{\theta}, \boldsymbol{w})$ the set of all ergodic occupation measures for the prescribed $\boldsymbol{\theta}$ and $\boldsymbol{w}$ on state-action space $S^{(\theta)} \times U^{(\theta)}$ for the controlled Markov process $\boldsymbol{Z}^{(\theta)}$ with policy $\pi^{(\theta)}$. Define

$$\tilde{\boldsymbol{g}}(\boldsymbol{\theta}, \boldsymbol{w}, \boldsymbol{\nu}) = \int \boldsymbol{g}(\boldsymbol{\theta}, \boldsymbol{w}, \boldsymbol{z}) \, \boldsymbol{\nu}(\mathrm{d}\boldsymbol{z}, U^{(w)}) \quad (60)$$

for $\boldsymbol{\nu}$ a measure on $S^{(w)} \times U^{(w)}$ and the Marchaud map

$$\hat{\boldsymbol{g}}(\boldsymbol{\theta}, \boldsymbol{w}) = \{\tilde{\boldsymbol{g}}(\boldsymbol{\theta}, \boldsymbol{w}, \boldsymbol{\nu}) : \boldsymbol{\nu} \in D^{(w)}(\boldsymbol{\theta}, \boldsymbol{w})\} \,. \quad (61)$$

We assume that the set $D^{(w)}(\boldsymbol{\theta}, \boldsymbol{w})$ is singleton, that is, $\hat{\boldsymbol{g}}(\boldsymbol{\theta}, \boldsymbol{w})$ contains a single function and we use the same notation for the set and its single element. If the set is not a singleton, the assumption of a solution can be expressed by the differential inclusion $\dot{\boldsymbol{w}}(t) \in \hat{\boldsymbol{g}}(\boldsymbol{\theta}, \boldsymbol{w}(t))$ [12].

$\forall \boldsymbol{\theta} \in \mathbb{R}^m$, the ODE

$$\dot{\boldsymbol{w}}(t) = \hat{\boldsymbol{g}}(\boldsymbol{\theta}, \boldsymbol{w}(t)) \quad (62)$$

has an asymptotically stable equilibrium $\boldsymbol{\lambda}(\boldsymbol{\theta})$ with domain of attraction $G_\theta$ where $\boldsymbol{\lambda} : \mathbb{R}^m \rightarrow \mathbb{R}^k$ is a Lipschitz map with constant $K$. Moreover, the function $V : G \rightarrow [0, \infty)$ is continuously differentiable where $V(\boldsymbol{\theta}, .)$ is the Lyapunov function for $\boldsymbol{\lambda}(\boldsymbol{\theta})$ and $G = \{(\boldsymbol{\theta}, \boldsymbol{w}) : \boldsymbol{w} \in G_\theta, \boldsymbol{\theta} \in \mathbb{R}^m\}$. This extra condition is needed so that the set $\{(\boldsymbol{\theta}, \boldsymbol{\lambda}(\boldsymbol{\theta})) : \boldsymbol{\theta} \in \mathbb{R}^m\}$ becomes an asymptotically stable set of the coupled ODE

$$\dot{\boldsymbol{w}}(t) = \hat{\boldsymbol{g}}(\boldsymbol{\theta}(t), \boldsymbol{w}(t)) \quad (63)$$

$$\dot{\boldsymbol{\theta}}(t) = 0 \,. \quad (64)$$

**(A7)** Assumption of bounded iterates:

$$\sup_n \|\boldsymbol{\theta}_n\| < \infty \text{ a.s. }, \quad (65)$$

$$\sup_n \|\boldsymbol{w}_n\| < \infty \text{ a.s.} \quad (66)$$

**Convergence Theorem.** The following theorem is from Karmakar & Bhatnagar [12]:

**Theorem 3** (Karmakar & Bhatnagar). *Under above assumptions if for all $\boldsymbol{\theta} \in \mathbb{R}^m$, with probability 1, $\{\boldsymbol{w}_n\}$ belongs to a compact subset $Q_\theta$ (depending on the sample point) of $G_\theta$ "eventually", then*

$$(\boldsymbol{\theta}_n, \boldsymbol{w}_n) \rightarrow \cup_{\boldsymbol{\theta}^* \in A_0} (\boldsymbol{\theta}^*, \boldsymbol{\lambda}(\boldsymbol{\theta}^*)) \quad a.s. \quad as \ n \rightarrow \infty \,, \quad (67)$$

*where $A_0 = \cap_{t \geqslant 0} \overline{\{\bar{\boldsymbol{\theta}}(s) : s \geqslant t\}}$ which is almost everywhere an internally chain transitive set of the differential inclusion*

$$\dot{\boldsymbol{\theta}}(t) \in \hat{\boldsymbol{h}}(\boldsymbol{\theta}(t)), \quad (68)$$

*where $\hat{\boldsymbol{h}}(\boldsymbol{\theta}) = \{\tilde{\boldsymbol{h}}(\boldsymbol{\theta}, \boldsymbol{\lambda}(\boldsymbol{\theta}), \boldsymbol{\nu}) : \boldsymbol{\nu} \in D^{(w)}(\boldsymbol{\theta}, \boldsymbol{\lambda}(\boldsymbol{\theta}))\}$.*

**Comments.**

**(C1)** This framework allows to show convergence for gradient descent methods beyond stochastic gradient like for the ADAM procedure where current learning parameters are memorized and updated. The random processes $\boldsymbol{Z}^{(w)}$ and $\boldsymbol{Z}^{(\theta)}$ may track the current learning status for the fast and slow iterate, respectively.

**(C2)** Stochastic regularization like dropout is covered via the random processes $A^{(w)}$ and $A^{(\theta)}$.

## 2.2 Rate of Convergence of Two Time-Scale Stochastic Approximation Algorithms

### 2.2.1 Linear Update Rules

First we consider linear iterates according to the PhD thesis of Konda [14] and Konda & Tsitsiklis [17].

$$\boldsymbol{\theta}_{n+1} = \boldsymbol{\theta}_n + a(n)\left(\boldsymbol{a}_1 - \boldsymbol{A}_{11}\,\boldsymbol{\theta}_n - \boldsymbol{A}_{12}\,\boldsymbol{w}_n + \boldsymbol{M}_n^{(\theta)}\right), \tag{69}$$

$$\boldsymbol{w}_{n+1} = \boldsymbol{w}_n + b(n)\left(\boldsymbol{a}_2 - \boldsymbol{A}_{21}\,\boldsymbol{\theta}_n - \boldsymbol{A}_{22}\,\boldsymbol{w}_n + \boldsymbol{M}_n^{(w)}\right). \tag{70}$$

**Assumptions.** We make the following assumptions:

**(A1)** The random variables $(\boldsymbol{M}_n^{(\theta)}, \boldsymbol{M}_n^{(w)}), n = 0, 1, \ldots$, are independent of $\boldsymbol{w}_0, \boldsymbol{\theta}_0$ and of each other. The have zero mean: $\mathrm{E}[\boldsymbol{M}_n^{(\theta)}] = 0$ and $\mathrm{E}[\boldsymbol{M}_n^{(w)}] = 0$. The covariance is

$$\mathrm{E}\left[\boldsymbol{M}_n^{(\theta)}\,(\boldsymbol{M}_n^{(\theta)})^T\right] = \boldsymbol{\Gamma}_{11}, \tag{71}$$

$$\mathrm{E}\left[\boldsymbol{M}_n^{(\theta)}\,(\boldsymbol{M}_n^{(w)})^T\right] = \boldsymbol{\Gamma}_{12} = \boldsymbol{\Gamma}_{21}^T, \tag{72}$$

$$\mathrm{E}\left[\boldsymbol{M}_n^{(w)}\,(\boldsymbol{M}_n^{(w)})^T\right] = \boldsymbol{\Gamma}_{22}. \tag{73}$$

**(A2)** The learning rates are deterministic, positive, nondecreasing and satisfy with $\epsilon \leqslant 0$:

$$\sum_n a(n) = \infty \quad, \quad \lim_{n\to\infty} a(n) = 0, \tag{74}$$

$$\sum_n b(n) = \infty \quad, \quad \lim_{n\to\infty} b(n) = 0, \tag{75}$$

$$\frac{a(n)}{b(n)} \to \epsilon. \tag{76}$$

We often consider the case $\epsilon = 0$.

**(A3)** Convergence of the iterates: We define

$$\boldsymbol{\Delta} := \boldsymbol{A}_{11} - \boldsymbol{A}_{12}\boldsymbol{A}_{22}^{-1}\boldsymbol{A}_{21}. \tag{77}$$

A matrix is *Hurwitz* if the real part of each eigenvalue is strictly negative. We assume that the matrices $-\boldsymbol{A}_{22}$ and $-\boldsymbol{\Delta}$ are Hurwitz.

**(A4)** Convergence rate remains simple:

(a) There exists a constant $\bar{a} \leqslant 0$ such that

$$\lim_n (a(n+1)^{-1} - a(n)^{-1}) = \bar{a}. \tag{78}$$

(b) If $\epsilon = 0$, then

$$\lim_n (b(n+1)^{-1} - b(n)^{-1}) = 0. \tag{79}$$

(c) The matrix

$$-\left(\boldsymbol{\Delta} - \frac{\bar{a}}{2}\,\boldsymbol{I}\right) \tag{80}$$

is Hurwitz.

**Rate of Convergence Theorem.** The next theorem is taken from Konda [14] and Konda & Tsitsiklis [17].

Let $\boldsymbol{\theta}^* \in \mathbb{R}^m$ and $\boldsymbol{w}^* \in \mathbb{R}^k$ be the unique solution to the system of linear equations

$$A_{11} \, \boldsymbol{\theta}_n \, + \, A_{12} \, \boldsymbol{w}_n \, = \, \boldsymbol{a}_1 \, , \tag{81}$$

$$A_{21} \, \boldsymbol{\theta}_n \, + \, A_{22} \, \boldsymbol{w}_n \, = \, \boldsymbol{a}_2 \, . \tag{82}$$

For each $n$, let

$$\hat{\boldsymbol{\theta}}_n \, = \, \boldsymbol{\theta}_n \, - \, \boldsymbol{\theta}^* \, , \tag{83}$$

$$\hat{\boldsymbol{w}}_n \, = \, \boldsymbol{w}_n \, - \, A_{22}^{-1} \, (\boldsymbol{a}_2 \, - \, A_{21} \, \boldsymbol{\theta}_n) \, , \tag{84}$$

$$\boldsymbol{\Sigma}_{11}^n \, = \, \boldsymbol{\theta}_n^{-1} \, \mathrm{E}\left[ \hat{\boldsymbol{\theta}}_n \hat{\boldsymbol{\theta}}_n^T \right] \, , \tag{85}$$

$$\boldsymbol{\Sigma}_{12}^n \, = \, \left( \boldsymbol{\Sigma}_{21}^n \right)^T \, = \, \boldsymbol{\theta}_n^{-1} \, \mathrm{E}\left[ \hat{\boldsymbol{\theta}}_n \hat{\boldsymbol{w}}_n^T \right] \, , \tag{86}$$

$$\boldsymbol{\Sigma}_{22}^n \, = \, \boldsymbol{w}_n^{-1} \, \mathrm{E}\left[ \hat{\boldsymbol{w}}_n \hat{\boldsymbol{w}}_n^T \right] \, , \tag{87}$$

$$\boldsymbol{\Sigma}^n \, = \, \begin{pmatrix} \boldsymbol{\Sigma}_{11}^n & \boldsymbol{\Sigma}_{12}^n \\ \boldsymbol{\Sigma}_{21}^n & \boldsymbol{\Sigma}_{22}^n \end{pmatrix} \, . \tag{88}$$

**Theorem 4** (Konda & Tsitsiklis). *Under above assumptions and when the constant $\epsilon$ is sufficiently small, the limit matrices*

$$\boldsymbol{\Sigma}_{11}^{(\epsilon)} \, = \, \lim_n \boldsymbol{\Sigma}_{11}^n \, , \quad \boldsymbol{\Sigma}_{12}^{(\epsilon)} \, = \, \lim_n \boldsymbol{\Sigma}_{12}^n \, , \quad \boldsymbol{\Sigma}_{22}^{(\epsilon)} \, = \, \lim_n \boldsymbol{\Sigma}_{22}^n \, . \tag{89}$$

*exist. Furthermore, the matrix*

$$\boldsymbol{\Sigma}^{(0)} \, = \, \begin{pmatrix} \boldsymbol{\Sigma}_{11}^{(0)} & \boldsymbol{\Sigma}_{12}^{(0)} \\ \boldsymbol{\Sigma}_{21}^{(0)} & \boldsymbol{\Sigma}_{22}^{(0)} \end{pmatrix} \tag{90}$$

*is the unique solution to the following system of equations*

$$\boldsymbol{\Delta} \, \boldsymbol{\Sigma}_{11}^{(0)} \, + \, \boldsymbol{\Sigma}_{11}^{(0)} \, \boldsymbol{\Delta}^T \, - \, \bar{a} \, \boldsymbol{\Sigma}_{11}^{(0)} \, + \, A_{12} \, \boldsymbol{\Sigma}_{21}^{(0)} \, + \, \boldsymbol{\Sigma}_{12}^{(0)} \, A_{12}^T \, = \, \boldsymbol{\Gamma}_{11} \, , \tag{91}$$

$$A_{12} \, \boldsymbol{\Sigma}_{22}^{(0)} \, + \, \boldsymbol{\Sigma}_{12}^{(0)} \, A_{22}^T \, = \, \boldsymbol{\Gamma}_{12} \, , \tag{92}$$

$$A_{22} \, \boldsymbol{\Sigma}_{22}^{(0)} \, + \, \boldsymbol{\Sigma}_{22}^{(0)} \, A_{22}^T \, = \, \boldsymbol{\Gamma}_{22} \, . \tag{93}$$

*Finally,*

$$\lim_{\epsilon \downarrow 0} \boldsymbol{\Sigma}_{11}^{(\epsilon)} \, = \, \boldsymbol{\Sigma}_{11}^{(0)} \, , \quad \lim_{\epsilon \downarrow 0} \boldsymbol{\Sigma}_{12}^{(\epsilon)} \, = \, \boldsymbol{\Sigma}_{12}^{(0)} \, , \quad \lim_{\epsilon \downarrow 0} \boldsymbol{\Sigma}_{22}^{(\epsilon)} \, = \, \boldsymbol{\Sigma}_{22}^{(0)} \, . \tag{94}$$

The next theorems shows that the asymptotic covariance matrix of $a(n)^{-1/2} \boldsymbol{\theta}_n$ is the same as that of $a(n)^{-1/2} \bar{\boldsymbol{\theta}}_n$, where $\bar{\boldsymbol{\theta}}_n$ evolves according to the single time-scale stochastic iteration:

$$\bar{\boldsymbol{\theta}}_{n+1} \, = \, \bar{\boldsymbol{\theta}}_n \, + \, a(n) \, \left( \boldsymbol{a}_1 \, - \, A_{11} \, \bar{\boldsymbol{\theta}}_n \, - \, A_{12} \, \bar{\boldsymbol{w}}_n \, + \, M_n^{(\theta)} \right) \, , \tag{95}$$

$$\boldsymbol{0} \, = \, \boldsymbol{a}_2 \, - \, A_{21} \, \bar{\boldsymbol{\theta}}_n \, - \, A_{22} \, \bar{\boldsymbol{w}}_n \, + \, M_n^{(w)} \, . \tag{96}$$

The next theorem combines Theorem 2.8 of Konda & Tsitsiklis and Theorem 4.1 of Konda & Tsitsiklis:

**Theorem 5** (Konda & Tsitsiklis 2nd). *Under above assumptions*

$$\boldsymbol{\Sigma}_{11}^{(0)} \, = \, \lim_n a(n)^{-1} \, \mathrm{E}\left[ \bar{\boldsymbol{\theta}}_n \bar{\boldsymbol{\theta}}_n^T \right] \, . \tag{97}$$

*If the assumptions hold with $\epsilon = 0$, then $a(n)^{-1/2} \hat{\boldsymbol{\theta}}_n$ converges in distribution to $\mathcal{N}(\boldsymbol{0}, \boldsymbol{\Sigma}_{11}^{(0)})$.*

**Comments.**

**(C1)** In his PhD thesis [14] Konda extended the analysis to the nonlinear case. Konda makes a linearization of the nonlinear function $\boldsymbol{h}$ and $\boldsymbol{g}$ with

$$\boldsymbol{A}_{11} = -\frac{\partial \boldsymbol{h}}{\partial \boldsymbol{\theta}}, \quad \boldsymbol{A}_{12} = -\frac{\partial \boldsymbol{h}}{\partial \boldsymbol{w}}, \quad \boldsymbol{A}_{21} = -\frac{\partial \boldsymbol{g}}{\partial \boldsymbol{\theta}}, \quad \boldsymbol{A}_{22} = -\frac{\partial \boldsymbol{g}}{\partial \boldsymbol{w}}. \tag{98}$$

There are additional errors due to linearization which have to be considered. However, only a sketch of a proof is provided but not a complete proof.

**(C2)** Theorem 4.1 of Konda & Tsitsiklis is important to generalize to the nonlinear case.

**(C3)** The convergence rate is governed by $\boldsymbol{A}_{22}$ for the fast and $\boldsymbol{\Delta}$ for the slow iterate. $\boldsymbol{\Delta}$ in turn is affected by the interaction effects captured by $\boldsymbol{A}_{21}$ and $\boldsymbol{A}_{12}$ together with the inverse of $\boldsymbol{A}_{22}$.

### 2.2.2 Nonlinear Update Rules

The rate of convergence for nonlinear update rules according to Mokkadem & Pelletier is considered [20].

The iterates are

$$\boldsymbol{\theta}_{n+1} = \boldsymbol{\theta}_n + a(n)\left(\boldsymbol{h}(\boldsymbol{\theta}_n, \boldsymbol{w}_n) + \boldsymbol{Z}_n^{(\theta)} + \boldsymbol{M}_n^{(\theta)}\right), \tag{99}$$

$$\boldsymbol{w}_{n+1} = \boldsymbol{w}_n + b(n)\left(\boldsymbol{g}(\boldsymbol{\theta}_n, \boldsymbol{w}_n) + \boldsymbol{Z}_n^{(w)} + \boldsymbol{M}_n^{(w)}\right). \tag{100}$$

with the increasing $\sigma$-fields

$$\mathcal{F}_n = \sigma(\boldsymbol{\theta}_l, \boldsymbol{w}_l, \boldsymbol{M}_l^{(\theta)}, \boldsymbol{M}_l^{(w)}, \boldsymbol{Z}_l^{(\theta)}, \boldsymbol{Z}_l^{(w)}, l \leqslant n), \ n \geqslant 0. \tag{101}$$

The terms $\boldsymbol{Z}_n^{(\theta)}$ and $\boldsymbol{Z}_n^{(w)}$ can be used to address the error through linearization, that is, the difference of the nonlinear functions to their linear approximation.

**Assumptions.** We make the following assumptions:

**(A1)** Convergence is ensured:

$$\lim_{n\to\infty} \boldsymbol{\theta}_n = \boldsymbol{\theta}^* \ \text{a.s.}, \tag{102}$$

$$\lim_{n\to\infty} \boldsymbol{w}_n = \boldsymbol{w}^* \ \text{a.s.}. \tag{103}$$

**(A2)** Linear approximation and Hurwitz:
There exists a neighborhood $\mathcal{U}$ of $(\boldsymbol{\theta}^*, \boldsymbol{w}^*)$ such that, for all $(\boldsymbol{\theta}, \boldsymbol{w}) \in \mathcal{U}$

$$\begin{pmatrix} \boldsymbol{h}(\boldsymbol{\theta}, \boldsymbol{w}) \\ \boldsymbol{g}(\boldsymbol{\theta}, \boldsymbol{w}) \end{pmatrix} = \begin{pmatrix} \boldsymbol{A}_{11} & \boldsymbol{A}_{12} \\ \boldsymbol{A}_{21} & \boldsymbol{A}_{22} \end{pmatrix} \begin{pmatrix} \boldsymbol{\theta} - \boldsymbol{\theta}^* \\ \boldsymbol{w} - \boldsymbol{w}^* \end{pmatrix} + \mathrm{O}\left(\left\| \begin{matrix} \boldsymbol{\theta} - \boldsymbol{\theta}^* \\ \boldsymbol{w} - \boldsymbol{w}^* \end{matrix} \right\|^2\right). \tag{104}$$

We define

$$\boldsymbol{\Delta} := \boldsymbol{A}_{11} - \boldsymbol{A}_{12}\boldsymbol{A}_{22}^{-1}\boldsymbol{A}_{21}. \tag{105}$$

A matrix is *Hurwitz* if the real part of each eigenvalue is strictly negative. We assume that the matrices $\boldsymbol{A}_{22}$ and $\boldsymbol{\Delta}$ are Hurwitz.

**(A3)** Assumptions on the learning rates:

$$a(n) = a_0 \, n^{-\alpha} \tag{106}$$

$$b(n) = b_0 \, n^{-\beta}, \tag{107}$$

where $a_0 > 0$ and $b_0 > 0$ and $1/2 < \beta < \alpha \leqslant 1$. If $\alpha = 1$, then $a_0 > 1/(2e_{\min})$ with $e_{\min}$ as the absolute value of the largest eigenvalue of $\boldsymbol{\Delta}$ (the eigenvalue closest to 0).

**(A4)** Assumptions on the noise and error:

(a) martingale difference sequences:

$$\mathrm{E}\left[M_{n+1}^{(\theta)} \mid \mathcal{F}_n\right] = 0 \ \text{a.s.}, \tag{108}$$

$$\mathrm{E}\left[M_{n+1}^{(w)} \mid \mathcal{F}_n\right] = 0 \ \text{a.s.}. \tag{109}$$

(b) existing second moments:

$$\lim_{n\to\infty} \mathrm{E}\left[\begin{pmatrix} M_{n+1}^{(\theta)} \\ M_{n+1}^{(w)} \end{pmatrix} \left((M_{n+1}^{(\theta)})^T \quad (M_{n+1}^{(w)})^T\right) \mid \mathcal{F}_n\right] = \boldsymbol{\Gamma} = \begin{pmatrix} \boldsymbol{\Gamma}_{11} & \boldsymbol{\Gamma}_{12} \\ \boldsymbol{\Gamma}_{21} & \boldsymbol{\Gamma}_{22} \end{pmatrix} \ \text{a.s.} \tag{110}$$

(c) bounded moments:
There exist $l > 2/\beta$ such that

$$\sup_n \mathrm{E}\left[\|M_{n+1}^{(\theta)}\|^l \mid \mathcal{F}_n\right] < \infty \ \text{a.s.}, \tag{111}$$

$$\sup_n \mathrm{E}\left[\|M_{n+1}^{(w)}\|^l \mid \mathcal{F}_n\right] < \infty \ \text{a.s.} \tag{112}$$

(d) bounded error:

$$Z_n^{(\theta)} = r_n^{(\theta)} + \mathrm{O}\left(\|\boldsymbol{\theta} - \boldsymbol{\theta}^*\|^2 + \|\boldsymbol{w} - \boldsymbol{w}^*\|^2\right), \tag{113}$$

$$Z_n^{(w)} = r_n^{(w)} + \mathrm{O}\left(\|\boldsymbol{\theta} - \boldsymbol{\theta}^*\|^2 + \|\boldsymbol{w} - \boldsymbol{w}^*\|^2\right), \tag{114}$$

with

$$\|r_n^{(\theta)}\| + \|r_n^{(w)}\| = \mathrm{o}(\sqrt{a(n)}) \ \text{a.s.} \tag{115}$$

**Rate of Convergence Theorem.** We report a theorem and a proposition from Mokkadem & Pelletier [20]. However, first we have to define the covariance matrices $\boldsymbol{\Sigma}_\theta$ and $\boldsymbol{\Sigma}_w$ which govern the rate of convergence.

First we define

$$\boldsymbol{\Gamma}_\theta := \lim_{n\to\infty} \mathrm{E}\left[\left(M_{n+1}^{(\theta)} - \boldsymbol{A}_{12}\,\boldsymbol{A}_{22}^{-1}\,M_{n+1}^{(w)}\right)\left(M_{n+1}^{(\theta)} - \boldsymbol{A}_{12}\,\boldsymbol{A}_{22}^{-1}\,M_{n+1}^{(w)}\right)^T \mid \mathcal{F}_n\right] = \tag{116}$$

$$\boldsymbol{\Gamma}_{11} + \boldsymbol{A}_{12}\,\boldsymbol{A}_{22}^{-1}\,\boldsymbol{\Gamma}_{22}\,(\boldsymbol{A}_{22}^{-1})^T\,\boldsymbol{A}_{12}^T - \boldsymbol{\Gamma}_{12}(\boldsymbol{A}_{22}^{-1})^T\,\boldsymbol{A}_{12}^T - \boldsymbol{A}_{12}\,\boldsymbol{A}_{22}^{-1}\,\boldsymbol{\Gamma}_{21}\,.$$

We now define the asymptotic covariance matrices $\boldsymbol{\Sigma}_\theta$ and $\boldsymbol{\Sigma}_w$:

$$\boldsymbol{\Sigma}_\theta = \int_0^\infty \exp\left(\left(\boldsymbol{\Delta} + \frac{\mathbb{1}_{a=1}}{2\,a_0}\,\boldsymbol{I}\right)t\right)\boldsymbol{\Gamma}_\theta\,\exp\left(\left(\boldsymbol{\Delta}^T + \frac{\mathbb{1}_{a=1}}{2\,a_0}\,\boldsymbol{I}\right)t\right)\mathrm{d}t\,, \tag{117}$$

$$\boldsymbol{\Sigma}_w = \int_0^\infty \exp\left(\boldsymbol{A}_{22}\,t\right)\boldsymbol{\Gamma}_{22}\,\exp\left(\boldsymbol{A}_{22}\,t\right)\mathrm{d}t\,. \tag{118}$$

$\boldsymbol{\Sigma}_\theta$ and $\boldsymbol{\Sigma}_w$ are solutions of the Lyapunov equations:

$$\left(\boldsymbol{\Delta} + \frac{\mathbb{1}_{a=1}}{2\,a_0}\,\boldsymbol{I}\right)\boldsymbol{\Sigma}_\theta + \boldsymbol{\Sigma}_\theta\left(\boldsymbol{\Delta}^T + \frac{\mathbb{1}_{a=1}}{2\,a_0}\,\boldsymbol{I}\right) = -\boldsymbol{\Gamma}_\theta\,, \tag{119}$$

$$\boldsymbol{A}_{22}\,\boldsymbol{\Sigma}_w + \boldsymbol{\Sigma}_w\,\boldsymbol{A}_{22}^T = -\boldsymbol{\Gamma}_{22}\,. \tag{120}$$

**Theorem 6** (Mokkadem & Pelletier: Joint weak convergence). *Under above assumptions:*

$$\begin{pmatrix} \sqrt{a(n)^{-1}}\,(\boldsymbol{\theta} - \boldsymbol{\theta}^*) \\ \sqrt{b(n)^{-1}}\,(\boldsymbol{w} - \boldsymbol{w}^*) \end{pmatrix} \xrightarrow{\mathcal{D}} \mathcal{N}\left(\boldsymbol{0}, \begin{pmatrix} \boldsymbol{\Sigma}_\theta & \boldsymbol{0} \\ \boldsymbol{0} & \boldsymbol{\Sigma}_w \end{pmatrix}\right)\,. \tag{121}$$

**Theorem 7** (Mokkadem & Pelletier: Strong convergence). *Under above assumptions:*

$$\|\boldsymbol{\theta} - \boldsymbol{\theta}^*\| = \mathrm{O}\left(\sqrt{a(n)\,\log\left(\sum_{l=1}^n a(l)\right)}\right) \ \text{a.s.}, \tag{122}$$

$$\|\boldsymbol{w} - \boldsymbol{w}^*\| = \mathrm{O}\left(\sqrt{b(n)\,\log\left(\sum_{l=1}^n b(l)\right)}\right) \ \text{a.s.} \tag{123}$$

**Comments.**

**(C1)** Besides the learning steps $a(n)$ and $b(n)$, the convergence rate is governed by $\boldsymbol{A}_{22}$ for the fast and $\boldsymbol{\Delta}$ for the slow iterate. $\boldsymbol{\Delta}$ in turn is affected by interaction effects which are captured by $\boldsymbol{A}_{21}$ and $\boldsymbol{A}_{12}$ together with the inverse of $\boldsymbol{A}_{22}$.

## 2.3 Equal Time-Scale Stochastic Approximation Algorithms

In this subsection we consider the case when the learning rates have equal time-scale.

### 2.3.1 Equal Time-Scale for Saddle Point Iterates

If equal time-scales assumed then the iterates revisit infinite often an environment of the solution [28]. In Zhang 2007, the functions of the iterates are the derivatives of a Lagrangian with respect to the dual and primal variables [28]. The iterates are

$$\boldsymbol{\theta}_{n+1} \;=\; \boldsymbol{\theta}_n \;+\; a(n)\,\left(\boldsymbol{h}\big(\boldsymbol{\theta}_n,\boldsymbol{w}_n\big) \;+\; \boldsymbol{Z}_n^{(\theta)} \;+\; \boldsymbol{M}_n^{(\theta)}\right) , \tag{124}$$

$$\boldsymbol{w}_{n+1} \;=\; \boldsymbol{w}_n \;+\; a(n)\,\left(\boldsymbol{g}\big(\boldsymbol{\theta}_n,\boldsymbol{w}_n\big) \;+\; \boldsymbol{Z}_n^{(w)} \;+\; \boldsymbol{M}_n^{(w)}\right) . \tag{125}$$

with the increasing $\sigma$-fields

$$\mathcal{F}_n \;=\; \sigma(\boldsymbol{\theta}_l,\boldsymbol{w}_l,\boldsymbol{M}_l^{(\theta)},\boldsymbol{M}_l^{(w)},\boldsymbol{Z}_l^{(\theta)},\boldsymbol{Z}_l^{(w)},l\leqslant n),\; n\geqslant 0 . \tag{126}$$

The terms $\boldsymbol{Z}_n^{(\theta)}$ and $\boldsymbol{Z}_n^{(w)}$ subsum biased estimation errors.

**Assumptions.**   We make the following assumptions:

**(A1)** Assumptions on update function: $\boldsymbol{h}$ and $\boldsymbol{g}$ are continuous, differentiable, and bounded. The Jacobians

$$\frac{\partial \boldsymbol{g}}{\partial \boldsymbol{w}} \quad \text{and} \quad \frac{\partial \boldsymbol{h}}{\partial \boldsymbol{\theta}} \tag{127}$$

are Hurwitz. A matrix is *Hurwitz* if the real part of each eigenvalue is strictly negative. This assumptions corresponds to the assumption in [28] that the Lagrangian is concave in $\boldsymbol{w}$ and convex in $\boldsymbol{\theta}$.

**(A2)** Assumptions on noise:

$\{\boldsymbol{M}_n^{(\theta)}\}$ and $\{\boldsymbol{M}_n^{(w)}\}$ are a martingale difference sequences w.r.t. the increasing $\sigma$-fields $\mathcal{F}_n$. Furthermore they are mutually independent.

Bounded second moment:

$$\mathrm{E}\left[\|\boldsymbol{M}_{n+1}^{(\theta)}\|^2 \mid \mathcal{F}_n\right] \;<\; \infty \;\text{ a.s. } , \tag{128}$$

$$\mathrm{E}\left[\|\boldsymbol{M}_{n+1}^{(w)}\|^2 \mid \mathcal{F}_n\right] \;<\; \infty \;\text{ a.s. } . \tag{129}$$

**(A3)** Assumptions on the learning rate:

$$a(n) \;>\; 0 \quad,\quad a(n) \;\to\; 0 \quad,\quad \sum_n a(n) \;=\; \infty \quad,\quad \sum_n a^2(n) \;<\; \infty . \tag{130}$$

**(A4)** Assumption on the biased error:

Boundedness:

$$\limsup_n \|\boldsymbol{Z}_n^{(\theta)}\| \;\leqslant\; \alpha^{(\theta)} \;\text{ a.s.} \tag{131}$$

$$\limsup_n \|\boldsymbol{Z}_n^{(w)}\| \;\leqslant\; \alpha^{(w)} \;\text{ a.s.} \tag{132}$$

**Theorem.**   Define the "contraction region" $A_\eta$ as follows:

$$A_\eta \;=\; \{(\boldsymbol{\theta},\boldsymbol{w}) : \alpha^{(\theta)} \geqslant \eta\,\|\boldsymbol{h}(\boldsymbol{\theta},\boldsymbol{w})\| \quad \text{or} \quad \alpha^{(w)} \geqslant \eta\,\|\boldsymbol{g}(\boldsymbol{\theta},\boldsymbol{w})\|,\; 0\leqslant\eta<1\} . \tag{133}$$

**Theorem 8** (Zhang)**.** *Under above assumptions the iterates return to $A_\eta$ infinitely often with probability one (a.s.).*

**Comments.**

**(C1)** The proof of the theorem in [28] does not use the saddle point condition and not the fact that the functions of the iterates are derivatives of the same function.

**(C2)** For the unbiased case, Zhang showed in Theorem 3.1 of [28] that the iterates converge. However, he used the saddle point condition of the Lagrangian. He considered iterates with functions that are the derivatives of a Lagrangian with respect to the dual and primal variables [28].

### 2.3.2 Equal Time Step for Actor-Critic Method

If equal time-scales assumed then the iterates revisit infinite often an environment of the solution of DiCastro & Meir [7]. The iterates of DiCastro & Meir are derived for actor-critic learning.

To present the actor-critic update iterates, we have to define some functions and terms. $\boldsymbol{\mu}(\boldsymbol{u} \mid \boldsymbol{x}, \boldsymbol{\theta})$ is the policy function parametrized by $\boldsymbol{\theta} \in \mathbb{R}^m$ with observations $\boldsymbol{x} \in \mathcal{X}$ and actions $\boldsymbol{u} \in \mathcal{U}$. A Markov chain given by $\mathrm{P}(\boldsymbol{y} \mid \boldsymbol{x}, \boldsymbol{u})$ gives the next observation $\boldsymbol{y}$ using the observation $\boldsymbol{x}$ and the action $\boldsymbol{u}$. In each state $\boldsymbol{x}$ the agent receives a reward $r(\boldsymbol{x})$.

The average reward per stage is for the recurrent state $\boldsymbol{x}^*$:

$$\tilde{\eta}(\boldsymbol{\theta}) \;=\; \lim_{T \to \infty} \mathrm{E}\left[\frac{1}{T} \sum_{n=0}^{T-1} r(\boldsymbol{x}_n) \mid \boldsymbol{x}_0 = \boldsymbol{x}^*, \boldsymbol{\theta}\right] \;. \tag{134}$$

The estimate of $\tilde{\eta}$ is denoted by $\eta$.

The differential value function is

$$\tilde{h}(\boldsymbol{x}, \boldsymbol{\theta}) \;=\; \mathrm{E}\left[\sum_{n=0}^{T-1} (r(\boldsymbol{x}_n) \;-\; \tilde{\eta}(\boldsymbol{\theta})) \mid \boldsymbol{x}_0 = \boldsymbol{x}, \boldsymbol{\theta}\right] \;. \tag{135}$$

The temporal difference is

$$\tilde{d}(\boldsymbol{x}, \boldsymbol{y}, \boldsymbol{\theta}) \;=\; r(\boldsymbol{x}) \;-\; \tilde{\eta}(\boldsymbol{\theta}) \;+\; \tilde{h}(\boldsymbol{y}, \boldsymbol{\theta}) \;-\; \tilde{h}(\boldsymbol{x}, \boldsymbol{\theta}) \;. \tag{136}$$

The estimate of $\tilde{d}$ is denoted by $d$.

The likelihood ratio derivative $\boldsymbol{\Psi} \in \mathbb{R}^m$ is

$$\boldsymbol{\Psi}(\boldsymbol{x}, \boldsymbol{u}, \boldsymbol{\theta}) \;=\; \frac{\nabla_\theta \boldsymbol{\mu}(\boldsymbol{u} \mid \boldsymbol{x}, \boldsymbol{\theta})}{\boldsymbol{\mu}(\boldsymbol{u} \mid \boldsymbol{x}, \boldsymbol{\theta})} \;. \tag{137}$$

The value function $\tilde{h}$ is approximated by

$$h(\boldsymbol{x}, \boldsymbol{w}) \;=\; \boldsymbol{\phi}(\boldsymbol{x})^T \, \boldsymbol{w} \;, \tag{138}$$

where $\boldsymbol{\phi}(\boldsymbol{x}) \in \mathbb{R}^k$. We define $\boldsymbol{\Phi} \in \mathbb{R}^{|\mathcal{X}| \times k}$

$$\boldsymbol{\Phi} \;=\; \begin{pmatrix} \phi_1(\boldsymbol{x}_1) & \phi_2(\boldsymbol{x}_1) & \ldots & \phi_k(\boldsymbol{x}_1) \\ \phi_1(\boldsymbol{x}_2) & \phi_2(\boldsymbol{x}_2) & \ldots & \phi_k(\boldsymbol{x}_2) \\ \vdots & \vdots & & \vdots \\ \phi_1(\boldsymbol{x}_{|\mathcal{X}|}) & \phi_2(\boldsymbol{x}_{|\mathcal{X}|}) & \ldots & \phi_k(\boldsymbol{x}_{|\mathcal{X}|}) \end{pmatrix} \tag{139}$$

and

$$h(\boldsymbol{w}) \;=\; \boldsymbol{\Phi} \, \boldsymbol{w} \;. \tag{140}$$

For TD($\lambda$) we have an eligibility trace:

$$e_n \;=\; \lambda \, e_{n-1} \;+\; \boldsymbol{\phi}(\boldsymbol{x}_n) \;. \tag{141}$$

We define the approximation error with optimal parameter $\boldsymbol{w}^*(\boldsymbol{\theta})$:

$$\epsilon_{\mathrm{app}}(\boldsymbol{\theta}) \;=\; \inf_{\boldsymbol{w} \in \mathbb{R}^k} \|\tilde{h}(\boldsymbol{\theta}) \;-\; \boldsymbol{\Phi} \, \boldsymbol{w}\|_{\pi(\boldsymbol{\theta})} \;=\; \|\tilde{h}(\boldsymbol{\theta}) \;-\; \boldsymbol{\Phi} \, \boldsymbol{w}^*(\boldsymbol{\theta})\|_{\pi(\boldsymbol{\theta})} \;, \tag{142}$$

where $\pi(\boldsymbol{\theta})$ is an projection operator into the span of $\boldsymbol{\Phi}\boldsymbol{w}$. We bound this error by

$$\epsilon_{\text{app}} = \sup_{\boldsymbol{\theta} \in \mathbb{R}^k} \epsilon_{\text{app}}(\boldsymbol{\theta}) \,. \tag{143}$$

We denoted by $\tilde{\eta}$, $\tilde{d}$, and $\tilde{h}$ the exact functions and used for their approximation $\eta$, $d$, and $h$, respectively. We have learning rate adjustments $\Gamma_\eta$ and $\Gamma_w$ for the critic.

The update rules are:
**Critic:**

$$\eta_{n+1} = \eta_n + a(n)\,\Gamma_\eta\,(r(\boldsymbol{x}_n) - \eta_n) \,, \tag{144}$$

$$h(\boldsymbol{x}, \boldsymbol{w}_n) = \boldsymbol{\phi}(\boldsymbol{x})^T \boldsymbol{w}_n \,, \tag{145}$$

$$d(\boldsymbol{x}_n, \boldsymbol{x}_{n+1}, \boldsymbol{w}_n) = r(\boldsymbol{x}_n) - \eta_n + h(\boldsymbol{x}_{n+1}, \boldsymbol{w}_n) - h(\boldsymbol{x}_n, \boldsymbol{w}_n) \,, \tag{146}$$

$$e_n = \lambda\,e_{n-1} + \boldsymbol{\phi}(\boldsymbol{x}_n) \,, \tag{147}$$

$$\boldsymbol{w}_{n+1} = \boldsymbol{w}_n + a(n)\,\Gamma_w\,d(\boldsymbol{x}_n, \boldsymbol{x}_{n+1}, \boldsymbol{w}_n)\,e_n \,. \tag{148}$$

**Actor:**

$$\boldsymbol{\theta}_{n+1} = \boldsymbol{\theta}_n + a(n)\,\boldsymbol{\Psi}(\boldsymbol{x}_n, \boldsymbol{u}_n, \boldsymbol{\theta}_n)\,d(\boldsymbol{x}_n, \boldsymbol{x}_{n+1}, \boldsymbol{w}_n) \,. \tag{149}$$

**Assumptions.**   We make the following assumptions:

**(A1)** Assumption on rewards:
   The rewards $\{r(\boldsymbol{x})\}_{\boldsymbol{x} \in \mathcal{X}}$ are uniformly bounded by a finite constant $B_r$.

**(A2)** Assumption on the Markov chain:
   Each Markov chain for each $\boldsymbol{\theta}$ is aperiodic, recurrent, and irreducible.

**(A3)** Assumptions on the policy function:
   The conditional probability function $\boldsymbol{\mu}(\boldsymbol{u} \mid \boldsymbol{x}, \boldsymbol{\theta})$ is twice differentiable. Moreover, there exist positive constants, $B_{\mu_1}$ and $B_{\mu_2}$, such that for all $\boldsymbol{x} \in \mathcal{X}$, $\boldsymbol{u} \in \mathcal{U}$, $\boldsymbol{\theta} \in \mathbb{R}^m$ and $1 \leqslant l_1, l_2 \leqslant m$ we have

$$\left\| \frac{\partial \boldsymbol{\mu}(\boldsymbol{u} \mid \boldsymbol{x}, \boldsymbol{\theta})}{\partial \boldsymbol{\theta}_l} \right\| \leqslant B_{\mu_1} \,, \quad \left\| \frac{\partial^2 \boldsymbol{\mu}(\boldsymbol{u} \mid \boldsymbol{x}, \boldsymbol{\theta})}{\partial \boldsymbol{\theta}_{l_1}\,\partial \boldsymbol{\theta}_{l_2}} \right\| \leqslant B_{\mu_2} \,. \tag{150}$$

**(A4)** Assumption on the likelihood ratio derivative:
   For all $\boldsymbol{x} \in \mathcal{X}$, $\boldsymbol{u} \in \mathcal{U}$, and $\boldsymbol{\theta} \in \mathbb{R}^m$, there exists a positive constant $B_\Psi$, such that

$$\|\boldsymbol{\Psi}(\boldsymbol{x}, \boldsymbol{u}, \boldsymbol{\theta})\|_2 \leqslant B_\Psi < \infty \,, \tag{151}$$

   where $\|.\|_2$ is the Euclidean $L_2$ norm.

**(A5)** Assumptions on the approximation space given by $\boldsymbol{\Phi}$:
   The columns of the matrix $\boldsymbol{\Phi}$ are independent, that is, the form a basis of dimension $k$. The norms of the columns vectors of the matrix $\boldsymbol{\Phi}$ are bounded above by 1, that is, $\|\boldsymbol{\phi}_l\|_2 \leqslant 1$ for $1 \leqslant l \leqslant k$.

**(A6)** Assumptions on the learning rate:

$$\sum_n a(n) = \infty \quad, \quad \sum_n a^2(n) < \infty \,. \tag{152}$$

**Theorem.**   The algorithm converged if $\nabla_\theta \tilde{\eta}(\boldsymbol{\theta}) = \boldsymbol{0}$, since the actor reached a stationary point where the updates are zero. We assume that $\|\nabla_\theta \tilde{\eta}(\boldsymbol{\theta})\|$ hints at how close we are to the convergence point.

The next theorem from DiCastro & Meir [7] implies that the trajectory visits a neighborhood of a local maximum infinitely often. Although it may leave the local vicinity of the maximum, it is guaranteed to return to it infinitely often.

**Theorem 9** (DiCastro & Meir). *Define*

$$B_{\nabla\tilde{\eta}} \;=\; \frac{B_{\Delta td1}}{\Gamma_w} \;+\; \frac{B_{\Delta td2}}{\Gamma_\eta} \;+\; B_{\Delta td3}\,\epsilon_{\mathrm{app}}\,, \tag{153}$$

*where $B_{\Delta td1}$, $B_{\Delta td2}$, and $B_{\Delta td3}$ are finite constants depending on the Markov decision process and the agent parameters.*

*Under above assumptions*

$$\lim_{t\to\infty}\inf\;\|\nabla_\theta\tilde{\eta}(\boldsymbol{\theta}_t)\| \;\leqslant\; B_{\nabla\tilde{\eta}}\,. \tag{154}$$

*The trajectory visits a neighborhood of a local maximum infinitely often.*

**Comments.**

- **(C1)** The larger the critic learning rates $\Gamma_w$ and $\Gamma_\eta$ are, the smaller is the region around the local maximum.
- **(C2)** The results are in agreement with those of Zhang 2007 [28].
- **(C3)** Even if the results are derived for a special actor-critic setting, they carry over to a more general setting of the iterates.

## 3 ADAM Optimization as Stochastic Heavy Ball with Friction

The Nesterov Accelerated Gradient Descent (NAGD) [21] has raised considerable interest due to its numerical simplicity and its low complexity. Previous to NAGD and its derived methods there was Polyak's Heavy Ball method [23]. The idea of the Heavy Ball is a ball that evolves over the graph of a function $f$ with damping (due to friction) and acceleration. Therefore, this second-order dynamical system can be described by the ODE for the Heavy Ball with Friction (HBF) [10]:

$$\ddot{\boldsymbol{\theta}}_t \;+\; a(t)\,\dot{\boldsymbol{\theta}}_t \;+\; \nabla f(\boldsymbol{\theta}_t) \;=\; \mathbf{0}\,, \tag{155}$$

where $a(n)$ is the damping coefficient with $a(n) = \frac{a}{n^\beta}$ for $\beta \in (0,1]$. This ODE is equivalent to the integro-differential equation

$$\dot{\boldsymbol{\theta}}_t \;=\; -\,\frac{1}{k(t)}\,\int_0^t h(s)\nabla f(\boldsymbol{\theta}_s)\mathrm{d}s\,, \tag{156}$$

where $k$ and $h$ are two memory functions related to $a(t)$. For polynomially memoried HBF we have $k(t) = t^{\alpha+1}$ and $h(t) = (\alpha+1)t^\alpha$ for some positive $\alpha$, and for exponentially memoried HBF we have $k(t) = \lambda\exp(\lambda\,t)$ and $h(t) = \exp(\lambda\,t)$. For the sum of the learning rates, we obtain

$$\sum_{l=1}^n a(l) \;=\; a \begin{cases} \ln(n) \;+\; \gamma \;+\; \frac{1}{2n} \;+\; \mathrm{O}\!\left(\frac{1}{n^2}\right) & \text{for } \beta = 1 \\ \frac{n^{1-\beta}}{1-\beta} & \text{for } \beta < 1 \end{cases}, \tag{157}$$

where $\gamma = 0.5772156649$ is the Euler-Mascheroni constant.

Gadat et al. derived a discrete and stochastic version of the HBF [10]:

$$\boldsymbol{\theta}_{n+1} \;=\; \boldsymbol{\theta}_n \;-\; a(n+1)\,\boldsymbol{m}_n \tag{158}$$
$$\boldsymbol{m}_{n+1} \;=\; \boldsymbol{m}_n \;+\; a(n+1)\,r(n)\,\big(\nabla f(\boldsymbol{\theta}_n) \;-\; \boldsymbol{m}_n\big) \;+\; a(n+1)\,r(n)\,\boldsymbol{M}_{n+1}\,,$$

where

$$r(n) \;=\; \begin{cases} r & \text{for exponentially memoried HBF} \\ \frac{r}{\sum_{l=1}^n a(l)} & \text{for polynomially memoried HBF} \end{cases}. \tag{159}$$

This recursion can be rewritten as

$$\boldsymbol{\theta}_{n+1} \;=\; \boldsymbol{\theta}_n \;-\; a(n+1)\,\boldsymbol{m}_n \tag{160}$$
$$\boldsymbol{m}_{n+1} \;=\; \big(1 \;-\; a(n+1)\,r(n)\big)\,\boldsymbol{m}_n \;+\; a(n+1)\,r(n)\,\big(\nabla f(\boldsymbol{\theta}_n) \;+\; \boldsymbol{M}_{n+1}\big)\,. \tag{161}$$

The recursion Eq. (160) is the first moment update of ADAM [13].

For the term $r(n)a(n)$ we obtain for the polynomial memory the approximations

$$r(n)\, a(n) \;\approx\; r \begin{cases} \frac{1}{n \, \log n} & \text{for } \beta = 1 \\ \frac{1 - \beta}{n} & \text{for } \beta < 1 \end{cases}, \tag{162}$$

Gadat et al. showed that the recursion Eq. (158) converges for functions with at most quadratic grow [10]. The authors mention that convergence can be proofed for functions $f$ that are $L$-smooth, that is, the gradient is $L$-Lipschitz.

Kingma et al. [13] state in Theorem 4.1 convergence of ADAM while assuming that $\beta_1$, the first moment running average coefficient, decays exponentially. Furthermore they assume that $\frac{\beta_1^2}{\sqrt{\beta_2}} < 1$ and the learning rate $\alpha_t$ decays with $\alpha_t = \frac{\alpha}{\sqrt{t}}$.

ADAM divides $\boldsymbol{m}_n$ of the recursion Eq. (160) by the bias-corrected second raw moment estimate. Since the bias-corrected second raw moment estimate changes slowly, we consider it as an error.

$$\frac{1}{\sqrt{v + \Delta v}} \;\approx\; \frac{1}{\sqrt{v}} \;-\; \frac{1}{2\, v \, \sqrt{v}} \, \Delta v \;+\; \mathrm{O}(\Delta v^2) \,. \tag{163}$$

ADAM assumes the second moment $\mathrm{E}\left[g^2\right]$ to be stationary with its approximation $v_n$:

$$v_n \;=\; \frac{1 - \beta_2}{1 - \beta_2^n} \sum_{l=1}^{n} \beta_2^{n-l} \, g_l^2 \,. \tag{164}$$

$$\Delta_n v_n \;=\; v_n \;-\; v_{n-1} \;=\; \frac{1 - \beta_2}{1 - \beta_2^n} \sum_{l=1}^{n} \beta_2^{n-l} \, g_l^2 \;-\; \frac{1 - \beta_2}{1 - \beta_2^{n-1}} \sum_{l=1}^{n-1} \beta_2^{n-l-1} \, g_l^2 \tag{165}$$

$$= \frac{1 - \beta_2}{1 - \beta_2^n} \, g_n^2 \;+\; \frac{\beta_2 \, (1 - \beta_2)}{1 - \beta_2^n} \sum_{l=1}^{n-1} \beta_2^{n-l-1} \, g_l^2 \;-\; \frac{1 - \beta_2}{1 - \beta_2^{n-1}} \sum_{l=1}^{n-1} \beta_2^{n-l-1} \, g_l^2$$

$$= \frac{1 - \beta_2}{1 - \beta_2^n} \left( g_n^2 \;+\; \left(\beta_2 - \frac{1 - \beta_2^n}{1 - \beta_2^{n-1}}\right) \sum_{l=1}^{n-1} \beta_2^{n-l-1} \, g_l^2 \right)$$

$$= \frac{1 - \beta_2}{1 - \beta_2^n} \left( g_n^2 \;-\; \frac{1 - \beta_2}{1 - \beta_2^{n-1}} \sum_{l=1}^{n-1} \beta_2^{n-l-1} \, g_l^2 \right) \,.$$

Therefore

$$\mathrm{E}\left[\Delta_n v_n\right] \;=\; \mathrm{E}\left[v_n \;-\; v_{n-1}\right] \;=\; \frac{1 - \beta_2}{1 - \beta_2^n} \left( \mathrm{E}\left[g^2\right] \;-\; \frac{1 - \beta_2}{1 - \beta_2^{n-1}} \sum_{l=1}^{n-1} \beta_2^{n-l-1} \, \mathrm{E}\left[g^2\right] \right) \tag{166}$$

$$= \frac{1 - \beta_2}{1 - \beta_2^n} \left( \mathrm{E}\left[g^2\right] \;-\; \mathrm{E}\left[g^2\right] \right) \;=\; 0 \,.$$

We are interested in the difference of actual stochastic $v_n$ to the true stationary $v$:

$$\Delta v_n \;=\; v_n \;-\; v \;=\; \frac{1 - \beta_2}{1 - \beta_2^n} \sum_{l=1}^{n} \beta_2^{n-l} \left( g_l^2 \;-\; v \right) \,. \tag{167}$$

For a stationary second moment of $\boldsymbol{m}_n$ and $\beta_2 = 1 - \alpha a(n + 1)r(n)$, we have $\Delta v_n \propto a(n + 1)r(n)$. We use a linear approximation to ADAM's second moment normalization $1/\sqrt{v + \Delta v_n} \approx 1/\sqrt{v} - 1/(2v\sqrt{v})\Delta v_n + \mathrm{O}(\Delta^2 v_n)$. If we set $\boldsymbol{M}_{n+1}^{(v)} = -(\boldsymbol{m}_n \Delta v_n)/(2v\sqrt{v}a(n + 1)r(n))$, then $\boldsymbol{m}_n/\sqrt{v_n} \approx \boldsymbol{m}_n/\sqrt{v} + a(n + 1)r(n)\boldsymbol{M}_{n+1}^{(v)}$ and $\mathrm{E}\left[\boldsymbol{M}_{n+1}^{(v)}\right] = 0$, since $\mathrm{E}\left[g_l^2 - v\right] = 0$. For a stationary second moment of $\boldsymbol{m}_n$, $\{\boldsymbol{M}_n^{(v)}\}$ is a martingale difference sequence with a bounded second moment. Therefore $\{\boldsymbol{M}_{n+1}^{(v)}\}$ can be subsumed into $\{\boldsymbol{M}_{n+1}\}$ in update rules Eq. (160). The factor $1/\sqrt{v}$ can be incorporated into $a(n + 1)$ and $r(n)$.

# 4  Experiments: Additional Information

## 4.1  WGAN-GP on Image Data.

Table 1: The performance of WGAN-GP trained with the original procedure and with TTUR on CIFAR-10 and LSUN Bedrooms. We compare the performance with respect to the FID at the optimal number of iterations during training and wall-clock time in minutes.

| dataset | method | b, a | iter | time(m) | FID | method | b = a | iter | time(m) | FID |
|---|---|---|---|---|---|---|---|---|---|---|
| CIFAR-10 | TTUR | 3e-4, 1e-4 | 168k | 700 | **24.8** | orig | 1e-4 | 53k | 800 | 29.3 |
| LSUN | TTUR | 3e-4, 1e-4 | 80k | 1900 | **9.5** | orig | 1e-4 | 23k | 2010 | 20.5 |

## 4.2  WGAN-GP on the One Billion Word Benchmark.

Table 2: Samples generated by WGAN-GP trained on fhe One Billion Word benchmark with TTUR (left) the original method (right).

```
Dry Hall Sitning tven the concer      No say that tent Franstal at Bra
There are court phinchs hasffort      Caulh Paphionars tven got corfle
He scores a supponied foutver il      Resumaly , braaky facting he at
Bartfol reportings ane the depor      On toipe also houd , aid of sole
Seu hid , it 's watter 's remold      When Barrysels commono toprel to
Later fasted the store the inste      The Moster suprr tent Elay diccu
Indiwezal deducated belenseous K      The new vebators are demases to
Starfers on Rbama 's all is lead      Many 's lore wockerssaow 2 2 ) A
Inverdick oper , caldawho 's non      Andly , has le wordd Uold steali
She said , five by theically rec      But be the firmoters is no 200 s
RichI , Learly said remain .''''      Jermueciored a noval wan 't mar
Reforded live for they were like      Onles that his boud-park , the g
The plane was git finally fuels       ISLUN , The crather wilh a them
The skip lifely will neek by the      Fow 22o2 surgeedeto , theirestra
SEW McHardy Berfect was luadingu      Make Sebages of intarmamates , a
But I pol rated Franclezt is the      Gullla " has cautaria Thoug ly t
```

Table 3: The performance of WGAN-GP trained with the original procedure and with TTUR on the One Billion Word Benchmark. We compare the performance with respect to the JSD at the optimal number of iterations and wall-clock time in minutes during training. WGAN-GP trained with TTUR exhibits consistently a better FID.

| n-gram | method | b, a | iter | time(m) | JSD | method | b = a | iter | time(m) | JSD |
|---|---|---|---|---|---|---|---|---|---|---|
| 4-gram | TTUR | 3e-4, 1e-4 | 98k | 1150 | **0.35** | orig | 1e-4 | 33k | 1040 | 0.38 |
| 6-gram | TTUR | 3e-4, 1e-4 | 100k | 1120 | **0.74** | orig | 1e-4 | 32k | 1070 | 0.77 |

## 4.3  BEGAN

The Boundary Equilibrium GAN (BEGAN) [3] maintains an equilibrium between the discriminator and generator loss (cf. Section 3.3 in [3])

$$\mathrm{E}[\mathcal{L}(G(\boldsymbol{z}))] = \gamma \mathrm{E}[\mathcal{L}(\boldsymbol{x})] \tag{168}$$

which, in turn, also leads to a fixed relation between the two gradients, therefore, a two time-scale update is not ensured by solely adjusting the learning rates. Indeed, for stable learning rates, we see no differences in the learning progress between orig and TTUR as depicted in Figure 6.

Figure 6: Mean, maximum and minimum FID over eight runs for BEGAN training on CelebA and LSUN Bedrooms. TTUR learning rates are given as pairs $(b, a)$ of discriminator learning rate $b$ and generator learning rate $a$: "TTUR $b$ $a$". **Left:** CelebA, starting at mini-batch 10k for better visualisation. **Right:** LSUN Bedrooms. Orig and TTUR behave similar. For BEGAN we cannot ensure TTUR by adjusting learning rates.

# 5 Discriminator vs. Generator Learning Rate

The convergence proof for learning GANs with TTUR assumes that the generator learning rate will eventually become small enough to ensure convergence of the discriminator learning. At some time point, the perturbations of the discriminator updates by updates of the generator parameters are sufficient small to assure that the discriminator converges. Crucial for discriminator convergence is the magnitude of the perturbations which the generator induces into the discriminator updates. These perturbations are not only determined by the generator learning rate but also by its loss function, current value of the loss function, optimization method, size of the error signals that reach the generator (vanishing or exploding gradient), complexity of generator's learning task, architecture of the generator, regularization, and others. Consequently, the size of generator learning rate does not solely determine how large the perturbations of the discriminator updates are but serve to modulate them. Thus, the generator learning rate may be much larger than the discriminator learning rate without inducing large perturbation into the discriminator learning.

Even the learning dynamics of the generator is different from the learning dynamics of the discriminator, though they both have the same learning rate. Figure 7 shows the loss of the generator and the discriminator for an experiment with DCGAN on CelebA, where the learning rate was 0.0005 for both the discriminator and the generator. However, the discriminator loss is decreasing while the generator loss is increasing. This example shows that the learning rate neither determines the perturbations nor the progress in learning for two coupled update rules. The choice of the learning rate for the generator should be independent from choice for the discriminator. Also the search ranges of discriminator and generator learning rates should be independent from each other, but adjusted to the corresponding architecture, task, etc.

Figure 7: The respective losses of the discriminator and the generator show the different learning dynamics of the two networks.

# 6 Used Software, Datasets, Pretrained Models, and Implementations

We used the following datasets to evaluate GANs: The Large-scale CelebFaces Attributes (CelebA) dataset, aligned and cropped [19], the training dataset of the bedrooms category of the large scale image database (LSUN) [27], the CIFAR-10 training dataset [18], the Street View House Numbers training dataset (SVHN) [22], and the One Billion Word Benchmark [6].

All experiments rely on the respective reference implementations for the corresponding GAN model. The software framework for our experiments was Tensorflow 1.3 [1, 2] and Python 3.6. We used following software, datasets and pretrained models:

- BEGAN in Tensorflow, `https://github.com/carpedm20/BEGAN-tensorflow`, Fixed random seeds removed. Accessed: 2017-05-30
- DCGAN in Tensorflow, `https://github.com/carpedm20/DCGAN-tensorflow`, Fixed random seeds removed. Accessed: 2017-04-03
- Improved Training of Wasserstein GANs, image model, `https://github.com/igul222/improved_wgan_training/blob/master/gan_64x64.py`, Accessed: 2017-06-12
- Improved Training of Wasserstein GANs, language model, `https://github.com/igul222/improved_wgan_training/blob/master/gan_language.py`, Accessed: 2017-06-12
- Inception-v3 pretrained, `http://download.tensorflow.org/models/image/imagenet/inception-2015-12-05.tgz`, Accessed: 2017-05-02

Implementations are available at

- `https://github.com/bioinf-jku/TTUR`