[Reviews · NeurIPS 2017]

Reviewer 1



Generative adversarial networks (GANs) are turning out to be a very important advance in machine learning. Algorithms for training GANs have difficulties with convergence. The paper proposes a two time-scale update rule (TTUR) which is shown (proven) to converge under certain assumptions. Specifically, it shows that GAN Adam updates with TTUR can be expressed as ordinary differential equations, and therefore can be proved to converge using a similar approach as in Borkar' 1997 work. The recommendation is to use two different update rules for generator and discriminator, with the latter being faster, in order to have convergence guarantees. ***Concerns: 1-Assumptions A1 to A6 and the following Theorem 1 are symmetric wrt h and g, thus one can assume it makes no difference which one of the networks (discriminator or generator) has the faster updates. A discussion on why this symmetry does not hold in practice, as pointed out by the authors in introduction, is necessary. 2-The assumptions used in the proof should be investigated in each of the provided experiments to make sure the proof is in fact useful, for example one of the key assumptions that a = o(b) is clearly not true in experiments (they use a = O(a) instead). 3-The less variation pointed out in Figure 2 are probably due to lower learning rate of the generator in TTUR compared to SGD, however since the learning rate values are not reported for Figure 2, we can not be sure. 4-In Figure 6, variance and/or confidence intervals should be reported. 5-The reported learning rate for TTUR appears to be only linearly faster for the discriminator (3 times faster in Figure 6, 1.25 times faster in Figure 3). This is not consistent with the assumption A2. Moreover, it is well known that a much faster discriminator for regular GANs undermines the generator training. However, according to Theorem 1, the system should converge. This point needs to be addressed. 6-There is insufficient previous/related work discussion, specifically about the extensive use of two time updates in dynamic systems, as well as the research on GAN convergence. 7-In general, theorems, specifically Theorem 2, are not well explained and elaborated on. The exposited ideas can become more clear and the paper more readable with a line or two of explanation in each case. In summary, while the proofs for convergence are noteworthy, a more detailed investigation and explanation of the effect of assumptions in each of the experiments, and how deviating from them can hurt the convergence in practice, will help clarify the limitations of the proof.

Reviewer 2



The authors of the submission propose a two time-scale update rule for GAN training, which is essentially about using two different learning rates (or learning rate schedules) for generator and discriminator. I see the two main contributions of the submission as follows: 1) a formal proof of GAN convergence under TTUR assumptions, and 2) introduction of FID (Frechet Inception Distance) score for more meaningful measurement of performance. Contribution 1) alone is providing a significant step toward the theoretical understanding of GAN training. Contribution 2) provides a useful tool for future evaluation of generative model performance, and its motivation is clear from Section 4. Some minor comments: - I'm not quite sure what I'm supposed to see in Figures 4 vs. 5, and they take up a lot of relative space in the submission. - How were the final learning rates chosen? I feel like I'm missing the description of a recipe, in case this goes beyond brute-force search.

Reviewer 3



The paper proposes a two time-scale update rule (TTUR) for GAN training, which is to use different learning rates for the discriminator and the generator. Under certain assumptions, TTUR allows the proof of convergence of GANs, and the results carry to Adam. To evaluate the effectiveness of TTUR, the paper additionally proposed Frechet Inception Distance (FID), i.e., the approximate Frechet distance in the code space defined by the Inception model. Empirically, by tuning the two learning rates carefully, TTUR is able to achieve faster and more stable training as measured by FID, with comparable sample quality given the same FID score. The practical technique introduced by the paper is relatively simple and the contribution lies in the theoretical analysis of the convergence under TTUR. The assumptions and theorem in Section 2 mostly follow [1], and a similar analysis has been seen in proving the convergence of actor-critic algorithms using two time-scale update in reinforcement learning [2]. The result for Adam includes some careful treatment for the second-moment statistics. The intuitive explanation from line 76 to line 81 is helpful for understanding but too concise. More elaboration may help readers to appreciate the difference between TTSU and equal time scale update, and why the TTSU makes the difference here. As for experiments, it’s nice to see that a single update for the critic in TTUR improves upon the 10 updates for the critic in the improved WGAN in terms of wall time. But it would be good if the progress in terms of the number of outer iterations were also provided since it gives a sense of the algorithmic difference between one update with a larger learning rate vs. more updates with a smaller learning rate. Overall, despite the technical similarity to some existing work, the paper gives a proof for the convergence of GAN training under TTUR including Adam, which is a good contribution to the theory of GAN and should be included in the literature. [1] V. S. Borkar. Stochastic approximation with two time scales. [2] Konda, Vijay R., and John N. Tsitsiklis. "Actor-critic algorithms."